# Historical Analysis of Rainfall-Triggered Rockfalls: the Case Study of the Disaster of the Ancient Hydrothermal Sclafani Spa (Madonie Mts., Northern-Central Sicily, Italy) in 1851

Antonio Contino[1,2], Patrizia Bova[2], Giuseppe Esposito[1,2], Ignazio Giuffré[2] and Salvatore Monteleone[1]

[1]Dipartimento di Scienze della Terra e del Mare (DiSTeM), University of Palermo, Via Archirafi 20/22, 90123 Palermo, Italy

[2]Accademia Mediterranea Euracea di Scienze, Lettere e Arti (AMESLA), Via Gregorio Ugdulena 62, 90018 Termini Imerese (Palermo), Italy.

*Correspondence to*: A. Contino (ntncontino@gmail.com)

**Abstract.** In 1851, the region of Sicily experienced many rainstorm-induced landslides. On 13 March 1851, a rainstorm brought about a severe rockfall disaster near the small town of Sclafani (Madonie Mountains, northern-central Sicily, Italy). Rocks detached from the carbonate crest of Mt. Sclafani (813 m above sea level) fell downslope, causing the collapse of the ancient hydrothermal spa (about 430 m above sea level) and burying it under their fragments. Fortunately, the event did not cause injuries or victims. Given its geological, geomorphological and tectonic features, the calcareous-dolomitic and carbonate-siliciclastic relief of Mt. Sclafani is extremely prone to landsliding. This study combines the findings from detailed geological and geomorphological field surveys and from a critical review of documentary data. A thorough analysis of documentary sources and historical maps made it possible to identify the location (previously unknown) of the ancient spa. The rockfall dynamics was traced back by comparing field reconnaissance data and documentary sources. The 1851 event reconstruction is an example of application of an integrated methodological approach, which can yield a propaedeutic, yet meaningful picture of a natural disaster, paving the way to further research (e.g. slope failure susceptibility, future land-use planning, protection of thermal springs and mitigation of the impact of similar disasters in this area). Indeed, the intensification of extreme weather events, caused by global warming induced by climate change, has increased the risk of recurrence of a catastrophic event, like the one of the ancient Sclafani spa, which is always impending.

**Keywords**: rockfall, 1851 rainstorm, thermal springs, ancient Sclafani spa, Madonie Mountains, region of Sicily.

## 1 Introduction

The study discussed in this paper focuses on the rainfall-triggered rockfall that hit the spa of the small town of Sclafani Bagni (called Sclafani before 1953), in the watershed of the northern Imera River (Madonie Mts., Sicily).

Gravity-induced hydrogeomorphic movements, called landslides (WP/WLI, 1990), depend on a combination of multiple controlling factors (Santini et al., 2009): i) predisposing factors (e.g. lithology, morphology, tectonics, weathering); ii) triggering factors (e.g. intensity, duration and spatial distribution of rainfall, earthquakes, changes in soil cover); and iii) accelerating factors (e.g. human activities altering the natural stability of slopes).

Rockfalls may be regarded as relatively "minor" landslides, since they are confined to the removal and fast downslope movement of distinct fragments (of different size) of fresh rock, usually detached from a steep rock wall or cliff (Carson and Kirkby, 1972; Selby, 1982).

The types of rock movements may be: toppling, falling, bouncing, rolling and sliding, accompanied by the

fragmentation of the rock mass upon its impact on the slope (Erismann and Abele, 2001).

A large-scale mass movement of rock material is defined as rock avalanche, rockfall avalanche or rockslide (e.g.
Evans and Hungr, 1993; Cruden and Varnes, 1996; Erismann and Abele, 2001; Guzzetti et al., 2004; Hungr et al., 2014).

In the scientific literature, rockfalls are reported to have various causal factors. However, in many cases, they originate from a combination of topographic, geological and climatological factors, as well as of gravitational forces and time (Dorren, 2003 and references therein).

The disaster of Sclafani was associated with a rainstorm that took place in the region of Sicily (central Mediterranean) in 1851. The collected documentary data concern the rainstorm of 12-13 March 1851, which struck the northern-central sector of the Island, i.e. the Madonie Mts. The intense rainfall triggered many landslides. After the maximum intensity of rainfall, recorded on 13 March 1851, a rockfall ravaged the Sclafani spa and its infrastructure. This event is very seldom quoted in studies and databases concerning Sicilian landslides. Only one publication specifically refers
to the Sclafani rockfall in 1851 (Aureli et al., 2008). The history of the rockfall hitting the ancient Sclafani spa, its causes and consequences have never been investigated until now.

The data presented in this paper provide a detailed reconstruction of the Sclafani rockfall, based on thorough geological and geomorphological surveys, collection of historical records and careful cross-checks with related sources.

In recent times, Hungr (2004) stressed the importance of historical evidence, "potentially more accurate" than geological evidence (proxy data), even if "limited to the length of the historical period, often little more than 100 years in much of the world". The catastrophic event of Sclafani, which happened over 150 years ago, constitutes an interesting and emblematic case study.

The paper underlines the crucial importance of documentary data analysis to reconstruct the circumstances of
landslide events that occurred in historical times, providing a significant methodological and scientific contribution of a pioneering nature.

## 2 Materials and Methods

### 2.1 The Rainstorm of 1851

On 11 March 1851, a WNW windstorm battered Sicily. The storm moved approximately ESE (Tacchini, 1868), unleashing catastrophic floods and severe sea-storms on 12-13 March.

The extreme weather event affected the northern region and its central and western sectors almost entirely (Fig. 1) and was accompanied by heavy rain (Anonymous, 1847-59; Anonymous, 1851, Anonymous, 1851-52; Benso et al.,
1851; Di Marzo, 1859; Possenti, 1865; Aureli et al., 2008 and references therein). This torrential rain triggered a large number of shallow landslides in mountainous or hilly areas and floods in coastal plains, producing substantial

economic losses. In many places, numerous roads (some of which of strategic relevance) were damaged or blocked. A severe flood struck the city of Palermo and its outskirts (12-13 March 1851), with huge economic damage but fortunately no casualties (Anonymous, 1851; Anonymous, 1851-52; Mortillaro, 1887). The high water level reached

1.60 m above road level in the urban area of Palermo (Tacchini, 1868; Cusimano et al., 1989). Difficulties of drainage caused extensive coastal lagoons to persist for many days (e.g. in the Palermo plain, see Anonymous, 1851-52; Mortillaro, 1887). At Termini Imerese a severe sea-storm provoked the sinking of some boats and the death of eight fishermen (Anonymous, 1851-52).

On 12-13 March 1851, the only official rain gauge station in Sicily (Osservatorio Astronomico di Palermo,

hereinafter OAP, 78.5 m above sea level) measured 140 mm in 48 h (see Tacchini, 1868) (90.6% of the total monthly average, i.e. 154.43 mm; 18% of the total yearly average). Unfortunately, as the hourly records of 1851 disappeared during the insurrection of 1860, peak rainfall was not accurately determined. However, rainfall had its peak intensity on 13 March 1851 (69 mm in 14 h, from 8:00 to 22:00, see Tacchini, 1868). Another non-official rain gauge station (Istituto Nautico di Palermo, hereinafter INP, 21.5 m a.s.l., see Tacchini, 1868) recorded 90.9 mm in 24 h (38% of the

total monthly average; 11.6% of the total yearly average). The monthly rainfall of March 1851 (239.03 mm), recorded at the NIP station, was the maximum of the month in the period from 1827 to 1866 (see Tacchini, 1868). In contrast, the monthly rainfall of March 1851, recorded by the OAP station, was 154.43 mm (see Tacchini, 1868). The INP, located by the sea (wharf of Palermo), at the foot of Mt. Pellegrino (600 m a.s.l.), reported higher precipitation values than the OAP; this finding is almost certainly due to the effect of local orography (see Tacchini, 1868).

Most of the landslide-induced damage occurred in central-northern Sicily (Madonie Mts., northern Imera and Torto river basins). The small towns of Montemaggiore (now Montemaggiore Belsito), Caltavuturo (Anonymous, 1847-59) and the ancient Sclafani spa suffered most of the damage (core disaster area). In south-western Sicily (Platani basin), the picturesque small town of Sutera suffered the destructive effects of some rockfalls (Benso et al., 1851).

The winter snowstorm and this torrential rain caused the drying-up of trees, the devastation of cropland, a prolonged

food scarcity and economic losses in many production activities (Pitré, 1871).

Prolonged torrential rains felled plants and trees, uprooted grapevines, initiated landslides and soil erosion, while streams spilled over their banks, covering grassland and grazing land with sand, gravel and rocks (Minà Palumbo, 1854). The perceived exceptionality of the event, recalled as "the deluge", was handed down from generation to generation through a dialectal folk song of Caltavuturo (Pitré, 1871). The locals were so impressed by the event that

they recollected as many as "forty days" of torrential rain, similarly to the Biblical Great Flood. They also defined the event as "the scourge of God" (Giornale Officiale di Sicilia, 1851), interpreting it as a divine punishment requiring prayers for the intercession of the Virgin Mary (Pitré, 1871).

Based on a general estimation, at least 26 buildings were destroyed, two bridges were damaged, eight fishermen drowned, cropland and crops were badly damaged (especially in the Termini Imerese and Cefalù districts, see Benso

et al., 1851). King Ferdinand II of Bourbon donated money to the people hit by the landslides and floods: 30,000

ducats, 12,000 of which only in the city of Palermo (Anonymous, 1851).

The rainstorm of 1851 may be classified among the Damaging Hydrogeological Events (DHE, see Petrucci and Polemio, 2003) due to its catastrophic effects and among the Very Large Catastrophic Events (VLCE, see Barriendos and Rodrigo, 2006) due to its extension to four or more basins.

1850-51 were rainy years (Tacchini, 1868). After a dry summer, the autumn of 1850 was extremely rainy. The monthly rainfall of September 1850 (138.57 mm) was the peak value of the month, in the period from 1827 to 1866, recorded by the non-official INP station (Tacchini, 1868). The 1850-51 winter, snowy and rainy, was followed by scarce rainfall in spring. This fact had phenological effects: insufficient wheat and olive production and extraordinary blooming of fruit-trees (e.g. apricot trees) and a species of Poaceae (*Ampelodesmos mauritanicum*), a typical example
of Mediterranean bunchgrass used for making strings and ropes, followed by limited production of foliage. In the summer, the anomalous proliferation of harmful *Coleoptera* destroyed many vineyards (Minà Palumbo, 1854).

### 2.2 Topographic, Geological and Tectonic Setting

Sclafani Bagni is a picturesque town of medieval age, just as its castle rising at the top of its slope (813 m a.s.l.); it is
located in central-northern Sicily (Madonie Mts.), about 80 km from Palermo (see Fig. S1 and S5 in Supplementary Information). In the past, its three hot sulfur springs (see Fig. S4 in Supplementary Information), lying at the foot of Mt. Sclafani and used for thermal balneology, were among the most famous in Sicily (Vinaj and Pinali, 1916-1923).

The history of the thermal baths of Sclafani may be divided into three phases. An original heterogeneous group of buildings was used for bathing and lodging (the so-called "thermal bath houses"). Some nearby houses were used for
additional lodging (the so-called "Giambelluca's houses", see Cacciatore, 1828; Cappa, 1847). In the years 1846-50, the first thermal establishment (the so-called "ancient Sclafani spa"), with two floors and 113 beds (Cappa, 1847), was built on the orders of the Hispanic nobleman Giuseppe Àlvarez de Toledo, Duke of Ferrandina, Count of Sclafani. This establishment, which included the original "thermal bath houses", was destroyed by the catastrophic rockfall of 1851 (see Di Marzo, 1859). After the disaster, the Count of Sclafani commissioned  the construction
(1856-57) of a new one-floor thermal establishment (the so-called "Masseria Bagni" i.e. "thermal bath farm")  in a site near the previous one, at an elevation of about 402 m a.s.l. (see Fig. S2 in Supplementary Information). The new thermal establishment, a unique example in the Madonie region,  featuring a new aqueduct and about 200 beds (Marieni, 1870) and covering an area of 1960 m$^2$, was totally abandoned after 1985.

The Madonie are the tallest mountains (1,979 m a.s.l. at Pizzo Carbonara) of the Sicilian northern chain. These reliefs,
a segment of the Sicilian Maghrebid chain, consist of stacked tectonic units, which originate from the deformation of distinct ancient paleogeographic domains, emplaced in Neogene times. The homonymous regional natural park was established (1989) in the Madonie Mts., given their high environmental value. The study area covers 82.7 km$^2$ it has an average altitude of 600 m a.s.l. and a maximum altitude of 1,794 m a.s.l. at Monte dei Cervi.

The main structure of the western Madonie Mts. is a Meso-Cenozoic sequence (Imerese and Sicilidi domains),

covered by syntectonic deposits (Catalano et al., 2011b, see Fig. 2).

The Imerese units derive from the deformation of a carbonate and carbonate-siliciclastic deep-sea succession (basin-slope, from the upper Triassic to the Oligocene, thickness 1,200-1,400 m, e. g. Basilone and Lo Cicero, 2009).

On Mt. Sclafani, the Imerese succession outcrops with a thickness of 600-900m and dips 10° to 80° obliquely to the slope. From bottom to top, the succession is made up of (Fig. 3): 1. grey marly limestones, thin-bedded and with chert nodules (Scillato fm., upper Triassic, exposed with a thickness of 100-130m); 2. four main bodies of massive white dolorudites, resedimented and alternating with thin-graded and -laminated doloarenites (Fanusi fm., lower Lias, thickness 230-250m); 3. thin-bedded variegated radiolarites; siliceous argillites with typical conjugate cleavage; toward base, marlstones and calcilutites body (thickness 10-20m, 3a) sandwiched (radiolarian member, Crisanti fm., middle Lias-Portlandian, thickness 150-250m); 4. resedimented grey calcareous breccias (Ellipsactinia breccias member, Portlandian-lower Cretaceous, Crisanti fm., thickness 50-80m); 5. reddish marls, jaspers, spiculites, interbedded with grey calcarenites (marly-spiculitic member, lower Cretaceous, Crisanti fm., thickness 40-100m); 6. resedimented grey calcareous breccias (Rudistid breccias member, upper Cretaceous, Crisanti fm., thickness 20-50m); 7. marlstones and marly limestones (Caltavuturo fm., Eocene-lower Oligocene, thickness 100-150m); 8. pelites and quartzarenites (Numidian Flysch, upper Oligocene-lower Miocene).

The basinal marine succession of the Sicilidi units (upper Cretaceous-lower Oligocene), detached from its original bedrock and tectonically overlying the Numidian flysch, consists of limbs of marly clays and marly limestones.

The structural edifice is unconformably overlain by a syntectonic wedge-top basin, filled with terrigenous, evaporitic and clastic-carbonate rocks (of Serravallian-Pliocene age). The most recent deposits include a local debris cover, colluvia and landslide deposits (Holocene).

During the Miocene and Pleistocene interval, two main tectonic events took place (Catalano et al., 2000): I) a first thrust system (currently SW-dipping), consisting in a quasi-horizontal, NE-SW-trending maximum compression (shallow-seated structures, middle Miocene); II) a more recent strike-slip and/or transpressional fault system, consisting in an N-S-trending maximum compression (deep-seated structures, after the beginning of the Tortonian).

The data about the tectonic architecture at the deep subsurface of the study area and nearby areas were obtained by interpreting the northern portion of the SI.RI.PRO ("SIsmica a RIflessione PROfonda", deep seismic reflection) borehole-calibrated seismic profile (Accaino et al., 2011; Catalano et al., 2013).

The upper portion of the SI.RI.PRO (Fig. 4) exhibits a stack of N-dipping, imbricated thrust sheets, affecting the thin basinal successions of Meso-Cenozoic age (Imerese unit, Numidian flysch and Sicilidi); the latter were emplaced during compressional event I. The Imerese thrust sheets appear to have been cut by deep-seated high-angle faults (generated at a deeper structural level) and then folded and lifted, giving rise to N-dipping features (i.e. backthrusts).

During the Pleistocene, NW-SE- and NE-SW-trending normal faults, correlated with an extensional tectonic event of a high angle, involved the study area. Finally, the occurrence of N-S- and E-W-oriented transtensional faults is consistent with a quasi-vertical maximum compression.

In the study area, N-S high-angle (about 70°) faults show an approximate rake of 60°. Kinematic indicators suggest a dextral transtensional movement.

### 2.3 Geomorphological and Morphostructural Setting

The geomorphological setting of the study area shows two different landscapes, each with well-defined landforms, depending on outcropping lithotypes and dominant morphogenetic processes: i) carbonate and carbonate-siliciclastic, with more or less steep slopes and numerous karst and morphoselection features; ii) clayey-marly with uneven and gently inclined slopes (<20°), usually affected by shallow gravitational movements (landslides). Alternating lithotypes with different erodibility and tectonic features justify the development of structural morphologies (cuestas, hogbacks and flatirons). Locally, NE-trending scarps display deep-seated gravitational deformations. N-S- and NE-SW-trending faults/fault lines, whose height was increased by selective erosion, dominate the landscape. On the ridge of the mountain (Cervi-Rocca di Sciara-Sclafani Anticlinorium, hereinafter CRSSA), the terrigenous covers (Numidian flysch) have been gradually dismantled since the salinity crisis of the Messinian, exposing calcareous-dolomitic and/or calcareous-siliciclastic levels to a different extent. The Plio-Quaternary uplift enhanced this process (Contino et al., 2015).

Fieldwork on Mt. Sclafani began with detailed geomorphological reconnaissance studies and mapping (original scale 1:10,000; area: 2.2 km$^2$, Fig. 5), based on the guidelines proposed by the Servizio Geologico Italiano (Gruppo di Lavoro per la Cartografia Geomorfologica, 1994). For general assessments of geomorphological mapping in geohazards, Lee (2001) recommends 1:10,000 as a suitable scale. The map of Fig. 5 is the synthesis of a meticulous field survey, which was fine-tuned by carefully interpreting topographical and cadastral maps, aerial photographs and satellite images.

Mt. Sclafani is a morphostructural high of the large CRSSA, markedly dissected by the recent faults with the highest angle. This typical isolated relief with a pyramidal shape and a triangular base is bordered on its E and W sides by an N-S-trending normal fault/fault line scarp. Its NW slope is bounded by a fault line scarp that was interpreted in different ways (see Discussion). Mt. Sclafani consists of three S-dipping slopes, separated by less steep slopes. Owing to selective erosion, this morphology corresponds to carbonate and siliciclastic levels, respectively. These levels give rise to fall-prone hard-on-soft landforms (see Fig. 3):

- Lower Cliff: about 230-10m high, calcareous-dolomitic (Scillato and Fanusi fms.); it shows selective erosion steps, with small caves and protruding ledges and is locally shaped by arcuate rockfall niches;

- Lower Talus Slope: roughly 240-40m high, siliceous and subordinately calcareous-marly (radiolarian member, Crisanti fm.), with morphoselection ledges;

- Middle Cliff: approximately 30-20m high, calcareous (Ellipsactinia breccias member, Crisanti fm.) and crowned by the ruins of the medieval castle; its steep undercut cliffs are rockfall-prone; this is the source area of the 1851 event (height approximately 730-750m a.s.l.);

- Middle Talus Slope: marly-calcareous (marly-spiculitic member, Crisanti fm.);

- Upper Cliff: calcareous (Rudistid breccias member, Crisanti fm.); also these steep undercut scarps are rockfall-prone;

- Upper Talus Slope: calcareous-marly (Caltavuturo fm.) and terrigenous (Numidian flysch).

### 2.4 Climate, Hydrography and Hydrogeology

Sicily has a typical Mediterranean climate with hot summers, drought and mild winters. By contrast, the Madonie Mts. have a cold climate in winter (December to February) and relatively mild summers. Average yearly temperature is roughly 14°C, whereas average yearly precipitation is about 800 mm. Precipitation is unevenly distributed over the different seasons. 85% of yearly precipitation is concentrated in September and April, whereas in summer (June-August) precipitation drops to less than 4% (Drago, 2002).

A relative peak occurs in September or October. On the highest reliefs (more than 1,000m a.s.l.), the yearly temperature excursion is equal to 10°C, while it decreases to 5-6° C in coastal areas.

The northern Imera river (29.10km; basinal area about 342km$^2$; maximum elevation: 1,869m a.s.l.) and its main left tributary, the Salito river (17km), drain the western sector of the Madonie Mts. At Scillato (basinal area 105km$^2$; hydrometric zero: 236m a.s.l., records available from 1976 to 1997 some missing data), the yearly average volume is 23.37Mm$^3$ and the yearly base flow volume is 7.83Mm$^3$. The maximum discharge (241.04m$^3$ s$^{-1}$) was measured during the severe rainstorm of 21 January 1981 (SOGESID, 2007).

The northern Imera river valley is open on its northern side, on the windward side of the northern Sicilian ridge; therefore, it is strongly exposed to the influence of the sea.

The monthly value of real evapotranspiration (ETR) was estimated with Thornthwaite's formula at 300 mm, based on the soil water balance for a field capacity of 20mm. Average surface discharge values (Ds) account for 24% of yearly precipitation (P) (see Drago, 2002)

The main aquifers of the Madonie Mts. are hosted in the calcareous-dolomitic (Scillato and Fanusi fms.) and calcareous (Crisanti fm.) rocks of the Imerese succession. The hydrostratigraphy of the Imerese succession, its structural geometries (ramps, faults and discontinuities) and karst landforms (of short length) govern groundwater flow.

The low-enthalpy thermo-mineral springs of Sclafani (33°-35°C; total flow value 6.7 l s$^{-1}$, e. g. Waring, 1965; Grassa et al., 2006 and references therein) are the superficial manifestation of a fault controlled geothermal system hosted in multilayered aquifers (carbonate and carbonate-siliciclastic units), locally confined beneath the terrigenous covers (Numidian Flysch).

### 2.5 Documentary Evidence

Written archives offer a large spectrum and an adequate resolution of evidence on past weather extremes and related

nature-induced disasters (Pfister, 2009 and references therein). Documentary evidence consists of different types of data: references; central and local public authorities reports and acts, engineers and experts synchronous relationships; ancient (Salemi, 1833; Massa, 1846-50) and recent series of maps (1930-2016), including satellite ones; synchronous engravings and drawings.

The historical reconstruction of the severe rainstorm of March 1851 and of the related Sclafani catastrophe was supported by three different types of evidence: i) direct description of the area of the thermal springs prior to the disaster by contemporary sources; these memories hold precious information about the landscape near the ancient thermal baths prior to the extreme event; ii) records of local and regional authorities concerning measures taken to respond to the terrible disaster (destruction of thermal baths, water mills, roads etc.), and iii) weather data kept by the meteorological station of the OAP (official) and by the INP (non-official). These records made it possible to confirm the exact day of the disaster (previously incorrectly reported), as well as the impact and magnitude of the rainstorm, i.e. the main triggering factor.

Therefore, documentary datasets were organised as follows: selected evidences previous to the disaster of the ancient hydrothermal Sclafani spa in 1851 (4 original documentary sources, see Supplementary Information, Table S1) and relatively to the disaster (22 original documentary sources, see Supplementary Information, Table S2), were collected in the first and second dataset, respectively. Evidences of the rainstorm of March 1851 was included in the third dataset (9 original documentary sources, see Supplementary Information, Table S3).

A significant legacy of documentary data, regarding the Sclafani event, is found in the pandects of the Bourbon administration in Palermo (Anonymous, 1849-60; Anonymous, 1847-59). The most important unpublished collection of synchronous documents (pandects), related to the event of March 1851, is kept in the State Archive of Palermo (Anonymous, 1847-59; Anonymous, 1849-60; Anonymous, 1851-52).

The collected historical documentary data were subjected to an essential critical analysis (homogenisation, cross-checking, validation and interpretation).

**3 Results and Discussion**

Geological and geomorphological mapping, field surveys and a critical analysis of documentary data considerably improved the understanding of the rockfall event in the study area. Unfortunately, no detailed eyewitness reports of the event were available.

In the Madonie Mts., very active infiltration processes, as well as weather and erosion agents, are significantly altering the bedrock, especially along structural discontinuities. The spacing between discontinuities has led to the development of unstable rock masses - which are prone to falling, especially during maximum intensity and/or duration of precipitation and under moisture-saturated freeze-thaw conditions - and to the formation of alluvial fans and talus deposits downslope (e.g. Scheidegger, 1975; Gardner, 1983; Whalley, 1984).

Given its geological, geomorphological and structural features, the calcareous-dolomitic and carbonate-siliciclastic

relief of Mt. Sclafani is extremely prone to rockfalls.

Regional tectonics plays a key role. In previous studies, the NW slope of Mt. Sclafani was interpreted as a tectonic line (Sclafani Fault, SF) with extensional kinematics (Agnesi et al., 2004; Catalano et al., 2011b; Gugliotta and Gasparo Morticelli, 2012), or as an indefinite fault/fault line scarp (Di Maggio et al., 1999). By contrast, in the study reported in this paper, the NW slope of Mt. Sclafani was interpreted as a transpressional fault line, juxtaposing lithotypes with different resistance to exogenous processes. The topographic expression of this transpressional line (Transpressional Sclafani Fault, hereinafter TSF), locally associated with normal faults, is marked by an abrupt scarp. This scarp, of arcuate shape and variable height, was shaped by the progressive collapse/retreat of the wall and by the incision of small fluviokarst canyons.

The nearby Rocca di Sciara Anticline (RSA), one of the structural highs of the Cervi-Rocca Sciara-Sclafani Anticlinorium (CRSSA), is cut by an SSE-dipping fault (Transpressional Cervi Fault, hereinafter TCF) with a high angle (roughly 70°) and an approximate rake of 15° to 45°. Kinematic indicators infer a sinistral transpressional movement (Gugliotta and Gasparo Morticelli, 2012).

Therefore, in this study (see structural sketch map in Fig, 2), the TSF was interpreted as the SW prolongation of the TCF. In agreement with these data, the SI.RI.PRO crustal seismic profile displays the buried Imerese tectonic high with N-dipping thrusts, which occurred in response to the basement thrust and associated uplift. The deep crustal uplift was ascribed to the Pliocene-lower Pleistocene (Catalano et al., 2013). The general uplift of this sector of the Madonie Mts. has accelerated the effects of exogenous processes and, consequently, gravitational movements.

In Mt. Sclafani, carbonate rock masses are segmented by discontinuity planes, intersected by a number of major vertical joints, which may increase the likelihood of future disasters.

Unfortunately, no published data are available about the level of fracturing of the bedrock and/or joints in the slopes of the study area, which are extremely steep and locally inaccessible or accessible only with mountaineering techniques. Hence, given the lack of reliable discontinuity data, no stability analysis was feasible. This topic will be discussed in a future publication.

Discontinuities in the carbonate rock have been altered by protracted exposure to rainwater. At the surface, the carbonate rock, divided into blocks, has given rise to ruiniform reliefs and trapezoidal spires.

During intense rainstorms (likely to induce water saturation and increase water pressure from precipitation-induced seepage), surface waters erode the weathered interspace materials.

Rockfalls associated with diffuse erosion and detrital slips occur along the edge of the carbonate layers that lie at the bottom and top of the site. These phenomena cause the toppling and rolling of carbonate blocks and rockfalls, especially as a result of intense rainstorms and under moisture-saturated freeze-thaw conditions. The same mechanism was observed to be at work throughout the area.

A detailed analysis of documentary evidence (including historical maps) made it possible to identify the exact position of the ancient Sclafani spa, which collapsed owing to and was buried by the 1851 rockfall.

In the 18<sup>th</sup> and early 19<sup>th</sup> centuries, the "thermal bath houses" were located at an elevation of about 430 m a.s.l., on a flat land (Mongitore 1743), near a small NW-SE-oriented valley, at the foot of the NW slope of Mt. Sclafani (Cacciatore, 1828). Before the event, the hydrothermal waters gushed out from the talus deposits at a distance of 30 m from the foot of a steep rock slope. An underground aqueduct conveyed water from the hot springs to the "thermal bath houses" (Cacciatore, 1828). The first improvements to the Sclafani baths lead back to the late 18<sup>th</sup> century-early

19<sup>th</sup> century (1792-99; 1825-27, see Campisi, 2015).

The ancient maps of the land surveyors Gaetano Salemi (1833) and Giuseppe Massa (1846-50) show the "thermal bath houses" and the "ancient Sclafani spa", respectively. The map of Salemi (Fig. 6), despite its naiveness, clearly displays the prominent spur at the base (Lower Cliff) of Mt. Sclafani, the position of the "thermal bath houses" and Giambelluca's houses, the thermal pond ("Gorga") and the four water mills. The map of Massa (Fig. 7) clearly shows

the location of the "ancient Sclafani spa" at the foot of the slope of Mt. Sclafani and the mule track connecting the small town with the thermal establishment.

The site of the ancient spa at the footslope of Mt. Sclafani is clearly visible in the pioneering geological and hydrogeological sketch (based on the Bourbon maps) drawn by Felice Giordano (Fig, 8), a mining engineer who worked in Sicily in 1860 (Daubrée, 1887). Furthermore, the  very detailed 1847 engraving (Cappa, 1847), portraying

the "ancient Sclafani spa" prior to the disaster, shows a site with a gentle topographic slope; this is due, in part, to the expansion work carried out at the time at the foot of the slope (Fig. 9).

The 1851 rockfalls, completely changed the landscape of the northern footslope of Mt. Sclafani (Fig. 10), and the position of the hydrothermal springs. Today, the site displays two mounds of heterogeneous rocky material, longitudinally sorted, with large rocks and masses (the largest ones are approximately equidimensional, 4 m in size),

at the foot of the lower cliff and a marked vegetational difference on the rockfall debris as against the surrounding areas. The morphological change took place on 13 March 1851 after the maximum intensity of rainfall, when the landslide ravaged the "ancient Sclafani hydrothermal spa". This event is very consistent with the data recorded by the two rain gauge stations of the city of Palermo (OAP and INP), indicating a rainfall peak on the same day.

The pandects of the Bourbon administrative office (Intendenza) of Palermo (Anonymous, 1847-59) documented that

the extraordinary torrential rains (the peak) battered the western Madonie on the night of 13 March 1851 and that streams swelled, provoking damages to structures (e.g. the aqueduct of Caltavuturo; see Supplementary Information, Table S2, source 13). Because of the landslides, the water spring called "Xhanimirici" (of unknown location) disappeared and the inhabitants of Sclafani remained without drinking water (Anonymous, 1849-60; see Supplementary Information, Table S2, sources 4-5).

Owing to an obvious misprint, Di Marzo (1859) reported this disaster on 19 March 1851 and this erroneous date was submissively repeated by many subsequent researchers.

Undoubtedly, failing eyewitness reports, documentary data do not permit to easily classify this historical disaster which took place over 150 years ago considering, among others, subsequent natural and anthropogenic changes (e.g., planting of tree species, terracing, excavations for road construction).

The disaster involved the catastrophic failure of carbonate rocks from the crest of the middle cliff (Ellipsactinia breccias), crowned by the ruins of a medieval castle. The event was a complex one: the type of initial failure evolved into another mechanism of movement, when the material advancing along the slope changed its volume, by incorporating materials entrained in its path (i. e. undisturbed rock from ledges or talus areas). The fragments continued their movement along the lower slope, hitting the edge of the lower cliff. Indeed, in the kinematics of the

event, the rockfall component cannot be ruled out, because the fragmented rock had to move beyond a break-away scarp (difference in height of about 70-90m; topographical gradient of about 50°-60°, see fig. 5) at the lower cliff. The material, probably bouncing or coming down along transport channels, fell downslope and reached the spa that consequently collapsed. The large building of the ancient spa, the access road, the nearby Giambelluca's houses, the banks of the thermal pond, the water mills and a segment of a mule track were completely destroyed and/or buried

under rock fragments. Conversely, the hot springs were unaffected, keeping the same chemical properties and the same discharge (Di Marzo, 1859). Fortunately, the disaster did not cause injuries or deaths because the thermal bathing season would start on the following 20 April (Vinaj and Pinali, 1916-1923). According to Jervis (1868), the event was particularly disastrous owing to  the unfavourable choice of the site of the ancient thermal spa.

The exceptional rainfall event of March 1851, which devastated this north-western area of the Madonie mountains,

must have certainly changed the lower talus slope (documentary sources report that the event caused an increase in ravines, see Supplementary Information, Table S2, source 14). This suggests that the morphology of the lower talus slope may have been extremely different from the current one, prior to the triggering of the Sclafani landslide. Hence, any attempt to reconstruct the trajectories of the 1851 event from the current morphology of the lower talus slope may reasonably be poorly reliable or unreliable.

In addition, the soft rocks (radiolarites and siliceous shales), which form the lower talus slope, are prone to erosion; in 150 years, they certainly experienced denudation and modelling processes (above all during extreme rainfall events: 1886, 1890, 1895, 1919, 1925, 1929, 1931, 1954, 1964, 1976-77; 1985, see Aureli et al. 2008) thus making any model useless. The synchronous engraving (see Fig. 9) representing the site of the ancient thermal spa shows the vegetation cover of the talus; this vegetation is supposed to have had an impact on the trajectories of fall of the

material. Unfortunately, Italian maps prior to the 20[th] century lack reliable indications on vegetation covers.

Moreover, the accumulated material does not reflect the composition of the lithotypes outcropping in the source area (Ellipsactinia breccias), but rather the one of the rocks present in the entire slope (Ellipsactinia breccias, radiolarites, siliceous shales, marls, calcilutites, dolomites etc.). The materials making up the landslide deposits are of variable size: very small for fragments of siliceous shales and/or radiolarites (cm/dm), progressively larger from calcilutites

(mostly dm, occasionally up to 1 m) to dolomites and limestones (from dm to some m). In the field, naturally-

exposed landslide deposits are often obliterated by a thick vegetal cover. Small partial exposures, resulting from earthworks, show carbonate blocks (limestones and dolomites), generally of metre scale, embedded into a chaotic mass of fragments of radiolarites, siliceous shales and slab-shaped calcilutites.

In addition, the morphology of the lower talus slope cannot be regarded as constant in time; therefore, empirical models are unable to predict the travel distance of future landslides (see Ayala-Carcedo et al., 2003) based on the data obtained for past events.

Geological and geomorphological evidence collected during field surveys, analyses of ancient maps, aerial and/or satellite images and historical data fit perfectly together, providing a detailed mapping of the area estimated (about 63,403 m$^2$) to be covered by the landslide deposits.

Some geometrical parameters of the Sclafani landslide could be determined: total height difference (height of fall) is about 385 m; horizontal distance (length of runout) is about 572 m; empirical shadow angle is about 31°-32°; ratio of H/L = 0.67.

Historical record collections do not include estimations of volume. Hence, a reliable estimation of the rock volume of the deposit and thickness is very difficult, because no pre-event topographic map, to be compared with subsequent surveys (e.g., official maps of Italy, 1878), is available. The official cartography of the Bourbon Kingdom, "Map of the Palermo Region" (scale 1:20,000; equidistance: 18.52 m; original survey of the "Topographic Office" in Naples: 1849-52) originally included the Sclafani section. Unfortunately, this section is missing in the cartographic archives of the Italian Military Geographical Institute (Florence).

An attempt to estimate the rock volume by using a new empirical relationship, proposed by Guzzetti et al. (2009), which links the surface area to the volume of the landslide. The resulting value was about 6.8 x 10$^5$ m$^3$. The same magnitude was obtained by using the volume vs. of H/L ratio graph (Tianchi, 1983).

Similarly, it was not possible to estimate the mass due to the heterogeneity of the deposit and the to difficulty of determining the percentages of its constituent materials.

In the synchronous documentary sources the landslide of Sclafani was called "scoscendimento" (see Supplementary Information, Table S2, sources 8, 10 and 12). Stoppani (1866) defined it as a "landslide of formidable proportions" with catastrophic effects, a veritable collapse of rock. In the Italian geological literature, the term "crollo di roccia" (rock fall) can be found only in the 19$^{th}$ century (see Almagià, 1910), but Gortani (1948) still uses "scoscendimento" or "rovina" (collapse). In view of this and considering that the surface covered by the accumulated material and the estimated volume are significant, it is reasonable to suppose that the type of initial failure was a rockslide, probably a "rock collapse" (sensu Hungr and Evans, 2004).

The catastrophic event was preceded by some premonitory signs around one year before. A resolution by the municipal council (so-called "decurionato") of Sclafani, adopted on 1 April 1850 (Anonymous, 1849-60; see Supplementary Information, Table S2, source 1), stated that a landslide had damaged the mule track connecting the small town with the thermal establishment. After the disaster, the source area (where the detachment surface has an

uneven pattern) continued to show signs of instability. The above-mentioned mule track was again damaged in January 1854 (based on the records, a detached rock blocked the passage, see Supplementary Information, Table S2, source 7).

In the 20[th] century, no rockfalls occurred during the serious weather event of 21-25 February 1931, but its effects may have been mitigated by the removal of part of the rocks (near the source area of the 1851 event) during the
construction of the road of access to the built-up centre, opened in 1930 (Termotto, 2009). This road, whose excavation required the use of explosives to cross the 1851 landslide rubble (now covered in part by vegetation), had a significant impact on the landscape, heavily changing its morphology, especially near the source area.

The severe weather events that occurred on 21 January 1981, 10-12 January 1987, 23-24 November 1991 and 13-14 November 1996 are likely to have reactivated landslides in Mt. Sclafani. During the 1990s, to mitigate the effects of
these events, the site was provided with a rockshed, as well as with nets and anchoring systems. Recently, eight active rockfall areas (one in the source area of the 1851 event) have been inventoried at Sclafani (Regione Siciliana, 2004).

**4 Conclusions**

Detailed field geological and geomorphological surveys in the study area highlighted multiple slope instabilities,
especially at shallow depth, such as the emblematic rockfall of 1851.

This study represents the starting point for a future Sicilian rockfall inventory database, a precious tool to shed more light onto these catastrophic events. The dynamics of the catastrophic rockfall involving the ancient Sclafani spa was reconstructed by analysing both historical and field data. The study combined thorough geological and geomorphological studies with the interpretation of aerial and satellite photos and was integrated with the analysis of
ancient and modern maps and a critical assessment of historical records. The manifold pieces of the Sclafani event puzzle, provided by documentary and geological evidence, fit entirely together, yielding a consistent picture of the impact of the disaster. The analysis of historical data i.e. that are the goals of the research study played a crucial role. The case study of Sclafani is an emblematic example that revives a catastrophic event so far ignored by the Italian inventories of landslide events (e.g. databases of ISPRA IFFI, AVI etc.).
The relevance of historical data in assessing the rockfall disaster was demonstrated by the investigation conducted as part of this case study.

According to some authors (e.g. Porter and Orombelli, 1980; Wieczorek and Jäger, 1996) detailed analyses of documentary data are crucial to identifying the mechanisms that trigger rockfalls, evaluating the susceptibility of the various slopes to rockfalls and developing magnitude-frequency relationships. Often, documentary data are the only
evidence of the socio-economic impacts of major disasters, such as rainstorms, severe floods and landslides, in periods preceding the deployment of instrumental monitoring networks (beginning in 1921 in Sicily).

The comparison of ancient and modern maps made it possible to determine the exact position of the ancient Sclafani spa, of paramount importance to carefully reconstruct the rockfall of 13 March 1851. The collected data are beginning

to provide a first picture of this catastrophic event.

The main causes of the rockfall disaster may be attributed to a number of intrinsic and extrinsic factors: geological, structural, geomorphological and, in particular, meteorological (extreme precipitation event). The main triggering factor of the Sclafani landslide was the exceptional rainfall event of 12-13 March 1851. There was a cause-effect relationship between the exceptional rainstorm and the landslide, as substantiated by the numerous historical data retrieved in this study (see Supplementary Information, Tables S2-S3). The area of Sclafani, typically mountainous, is

subject to freeze-thaw conditions. The earthquake events that produced macroseismic effects in the study area in the first half of the 19[th] century took place in 1818-19 and 1823 (Billi et al., 2010). Predisposing factors were many; some were intrinsic (related to the stratigraphic and tectonic setting), while other ones included selective erosion (hard-on-soft landforms). The anthropogenic impact changed the landscape near the source area (e.g. the road built in 1930, whose excavation required the use of explosives).

According to Zellmer (1987) the time, place and frequency of occurrence of rockfall disasters, as well as their scale, are unpredictable. However, some researchers are studying some possible precursors of rockfalls (mountain deformations: e.g. Bovis, 1990; seismic: Wang et al., 2003; Amitrano et al., 2005), including through monitoring systems (e.g. Schenato et al., 2013), in order to investigate the issue of prediction of these events, which are often catastrophic. The Sclafani rockfall of 1851 was an exceptional event that can hardly be framed in statistical terms.

In the 1986-2015 period, global warming induced by climate change expressed itself in different ways, including an increase of extreme precipitation and of catastrophic geomorphological processes (Katz and Brown, 1992; Rosenzweig et al., 2007; Sillmann and Roeckner, 2008). The higher frequency and intensity of rainfall activated landslides and/or debris flows (Evans and Clague, 1994; Keefer et al., 1987). High-impact events are expected to be more frequent and intense in the future, as a consequence of continuing global warming (Krautblatter and Moser,

2006; Field et al., 2012). Therefore, given the intensification of extreme weather events, the risk of recurrence of a catastrophic event, like the one of the ancient Sclafani spa, is always impending.

Future investigations based on direct and/or indirect methods (structural and geomechanical reconnaissance, seismic and/or microseismic monitoring, surveys via digital high-resolution photos, terrestrial and/or aerial laser scanning) may yield relatively new data to improve the assessment of rockfall susceptibility and adequately plan and manage

disaster prevention and mitigation actions in the Mt. Sclafani area (especially to safeguard the site of the thermal springs, which are particularly vulnerable). Unquestionably, a detailed study of past events is the starting point of any conceptual plan to mitigate rockfalls, as well as to design and develop systems for rock slope stabilisation. Finally, it is hoped that a future archaeological survey and excavation in the site of the ancient Sclafani spa can discover the remains of the collapsed establishment.


Data availability

The supporting datasets has not been deposited in a public repository being available as Supplementary Information files.

Team list

The correspondent author ensures that all authors are include in the authors list, its order has been agreed by all authors and that all authors are aware that the paper was submitted. The correspondent author declares: that has been authorized by the co-authors to enter into these arrangement; that the paper has not been published before and it is not under consideration for publication elsewhere; that the publication of the paper has been approved by all the author(s)

and by the responsible authorities – tacitly or explicitly – of the institutes where the work was carried out; that the authors have secured the right to reproduce any material; that has already been published or copyrighted elsewhere; that have agree to the Creative Commons Attribution 3.0 License.

Author contributions.

Author contributions are listed below. A. Contino developed ideas about the analysis and triggering mechanism of the Sclafani rockfall and contributed to the drafting of the paper together with P. Bova, G. Esposito and S. Monteleone; A. Contino, P. Bova, G. Esposito worked on the geological/geomorphological field mapping, collected and validated the documentary sources; A. Contino and I. Giuffré, made drawings. I. Giuffré edited the material and discussed the results and the interpretations with the other authors.


Conflict of Interest

The authors declare that they have no conflict of interest.

Disclaimer

The opinions, findings, and conclusions or recommendations expressed in the paper are those of the authors and do not necessarily reflect those of the NHESS or the organisation to which.

Acknowledgements

The authors acknowledge the Editorial Board Thomas Glade, the reviewers Alexander Preh and two anonymous

reviewer for their observations and comments of the manuscript. This work was financially supported by VIGOR project grants to S. Monteleone, DISTEM, University of Palermo, 2011-12. The authors would like to thank Belinda Butera (State Archive of Palermo, section of Catena), Rita Esposito (University of Palermo, Department of Geotechnics), Orazio Granata (Municipality of Sclafani Bagni), Anna Maria Lo Presti (Notarial Archive of Termini Imerese), Claudia Raimondo (Municipal Library of Termini Imerese), Filippo Picone (CRICD, Regione Siciliana,

Palermo), Nicolò Scalzo (State Archive of Palermo, section of Gancia), Maria Luisa Tarraga Baldó (Instituto de

Historia del Arte, CSIC, Consejo superior de investigaciones científicas, Madrid, Spain), Alberto Vitale (State Archive of Palermo, section of Termini Imerese) for their invaluable support during the search of documentary data. Stefania Saraceni is acknowledged for providing style correction.

Edited by: T. Glade
Reviewed by: Alexander Preh and two anonymous referees

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

Captions:

Figure 01 - Map of Sicily with the location of the study area and of the sites cited in the text. Map Datum World Geodetic System 1984 / Universal Transverse of Mercator Zone 32N.


Figure 02 - Simplified structural sketch of the study area showing the Cervi-Rocca Sciara-Sclafani Anticlinorium. SD (Syntectonic deposits); NF (Numidian Flysch, cap rock); SC (Sicilidi Units); IM (Imerese Units, multilayered reservoir). TSF (Transpressional Sclafani Fault); TCF (Transpressional Cervi Fault). Drawing modified after Catalano et al., 2011b, Gugliotta and Gasparo Morticelli, 2013.


Figure 03 – Schematic N-S geological and geomorphological profile (A-B) at the Mt. Sclafani. Details and explanation in the text, location of section in Fig. 5. Note the sequences of caprocks and soft rocks enhanced by selective weathering and erosion.

Figure 04 – Interpreted northern transect of the SI.RI.PRO. crustal seismic reflection profile, showing the hydrostructural setting of the tectonic units near the Sclafani area (drawing modified after Catalano et al., 2011a). NF (Numidian Flysch, cap rock); SC (Sicilidi Units); IM (Imerese Units, multilayered reservoir).

Figure 05 – Simplified geological and geomorphological map of Mt. Sclafani (equidistance of elevation contours:

20m), shown the sector crossed by the section (A-B) of Fig. 3 and the profile (c-d), of Fig. S6 (see Supplementary Information).

Figure 06 – Map of the fiefs and territory of Sclafani (detail) drawn by Gaetano Salemi (India ink and water colours, 1833; not oriented). The detail shows the small town, the prominent spur (brick floor in the map) at the base (Lower Cliff) of Mt. Sclafani, the "thermal bath houses" ("Bagni"), the thermal pond (number 17) and Ferrandina's water mills ("Molini"). State Archive of Palermo, section of Gancia; authorisation for reproduction no. 7/2016, ref. no. 5236 – 22 November 2016. The rhombus shows the location of a new one-floor thermal establishment (1856-57, the so-called "Masseria Bagni" i.e. "thermal bath farm").

Figure 07 – Sketch of the territory of Sclafani (detail) drawn by Giuseppe Massa (attrib. 1846-50; not in scale and without orientation). The detail shows the small town, the ancient Sclafani spa and the connecting mule track. Drawing modified after Caruso and Nobili, 2001.

Figure 08 – N-S geological and hydrogeological sketch (not in scale) of Mt. Sclafani drawn by Felice Giordano, approximately in 1860, using the Bourbon maps of 1849-52 (drawing modified after Doubrée, 1887). Legend: I. marls (Trias, Impermeable, not outcropping); Td. Dolomites (Trias); L. Limestones (Lias); Ca. clays (Cenomanian); Cc. Marls and limestones (upper Cretaceous); Ec. Limestones (lower Eocene); Ea. Speckled clays (middle Eocene); F. Fault; S. Thermal springs and location of the ancient Sclafani spa.

Figure 09 – Anonymous, very detailed engraving of the ancient Sclafani spa (from Cappa, 1847, courtesy of the Faculty of Medicine Library, Complutense University of Madrid). Note in the foreground a rock block (linked to more ancient rockfalls).

Figure 10 – The present landscape surrounding the site of the ancient Sclafani spa. The arrow indicates the location of the new thermal new one-floor thermal establishment (1856-57, the so-called "Masseria Bagni" i.e. "thermal bath farm"). The star indicates the today's location of Sclafani thermal springs. For vantage point, see Fig. S4 in Supplementary Information.

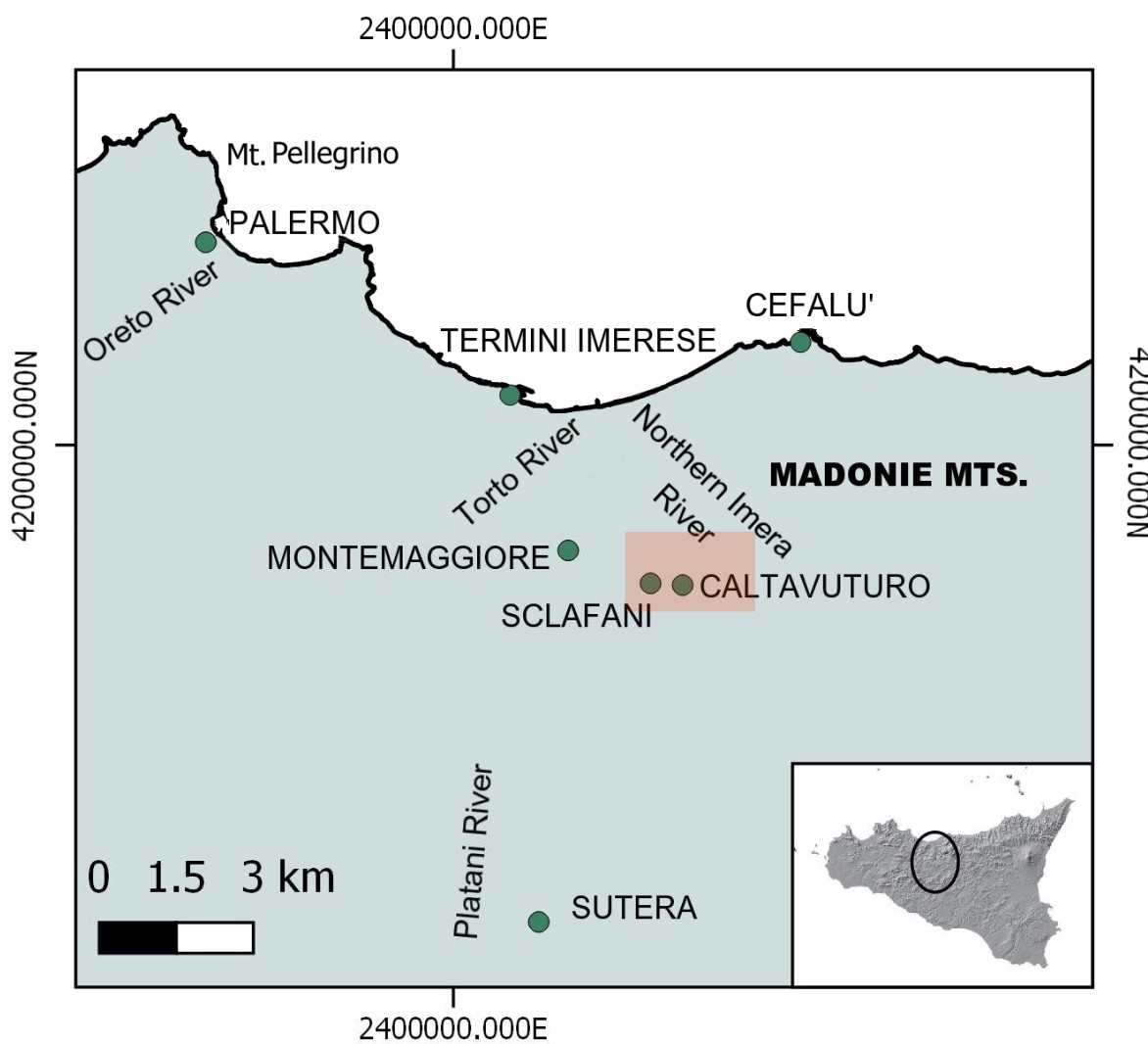

Fig. 01

745

750

755

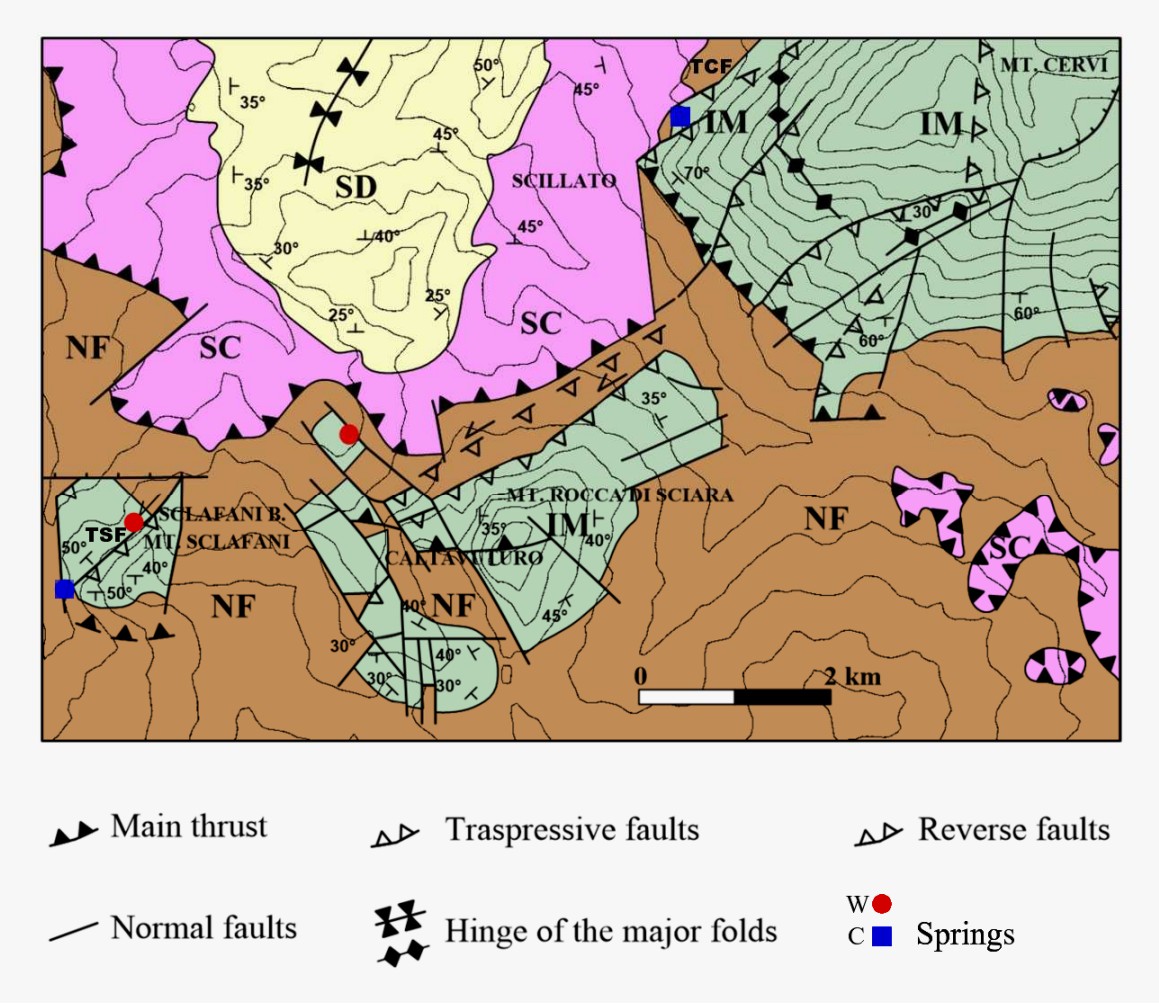

Fig. 02

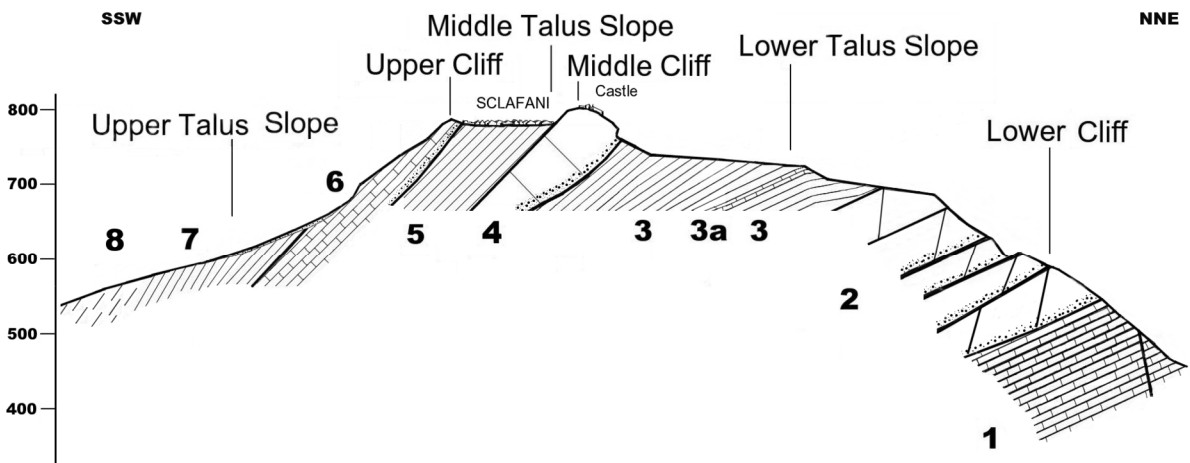

Fig. 03

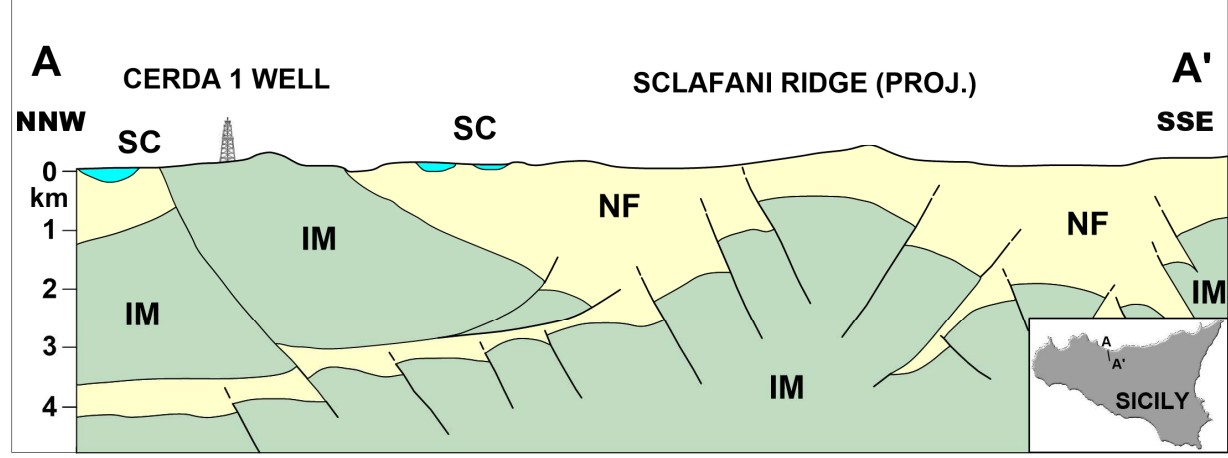

Fig. 04

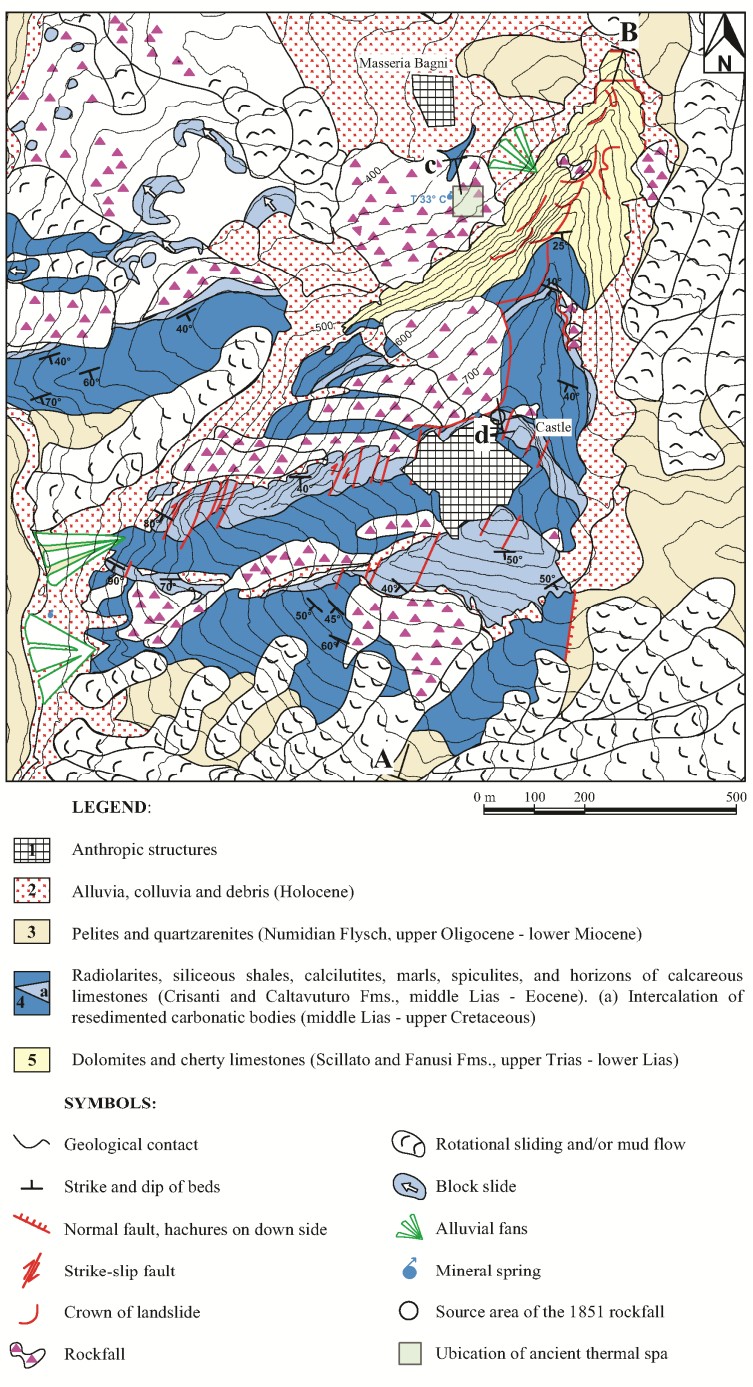

**LEGEND**:

| | |
|---|---|
| ▦ **1** | Anthropic structures |
| ⸬ **2** ⸬ | Alluvia, colluvia and debris (Holocene) |
| **3** | Pelites and quartzarenites (Numidian Flysch, upper Oligocene - lower Miocene) |
| **4** **a** | Radiolarites, siliceous shales, calcilutites, marls, spiculites, and horizons of calcareous limestones (Crisanti and Caltavuturo Fms., middle Lias - Eocene). (a) Intercalation of resedimented carbonatic bodies (middle Lias - upper Cretaceous) |
| **5** | Dolomites and cherty limestones (Scillato and Fanusi Fms., upper Trias - lower Lias) |

**SYMBOLS:**

| | | | |
|---|---|---|---|
| ⌣ | Geological contact | ◌ | Rotational sliding and/or mud flow |
| ⊥ | Strike and dip of beds | ◍ | Block slide |
| ⥤ | Normal fault, hachures on down side | ⋎ | Alluvial fans |
| ⚡ | Strike-slip fault | ● | Mineral spring |
| ⌐ | Crown of landslide | ○ | Source area of the 1851 rockfall |
| ◌ | Rockfall | ▢ | Ubication of ancient thermal spa |

Fig. 05

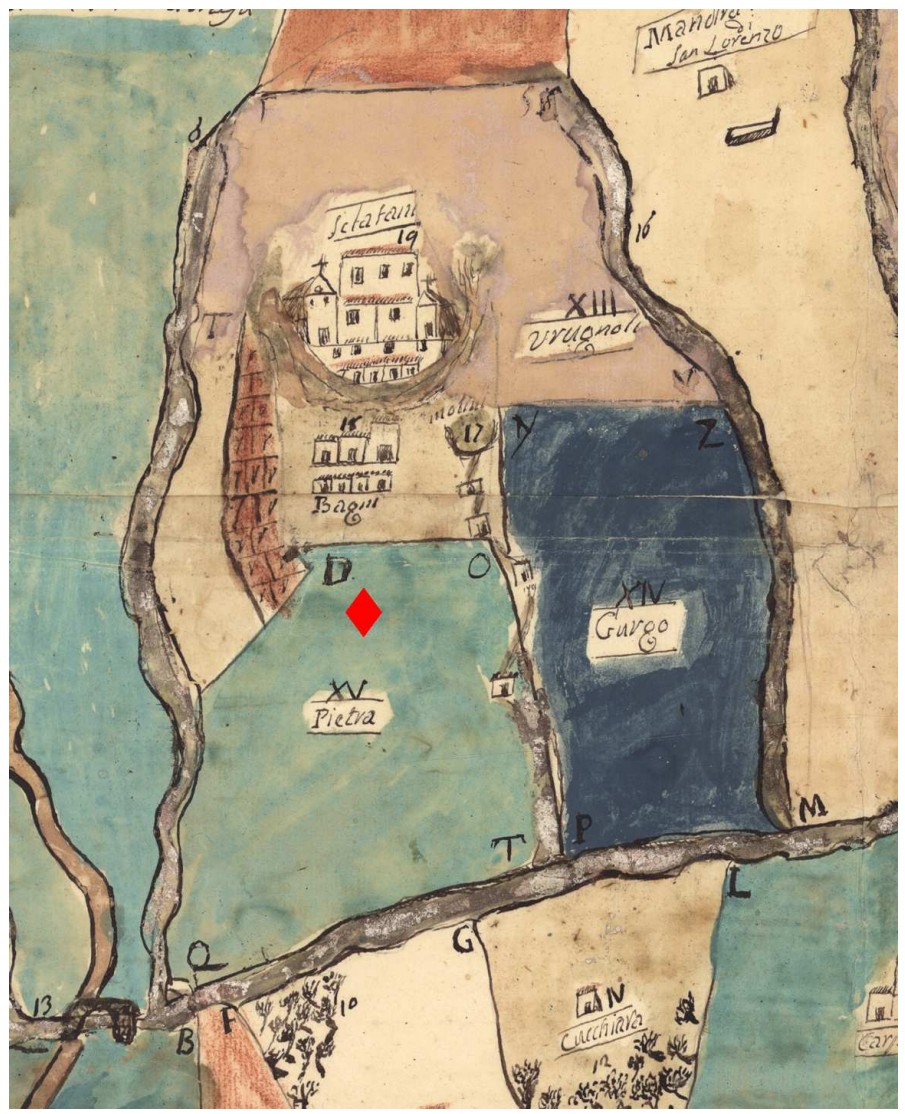

Fig. 06

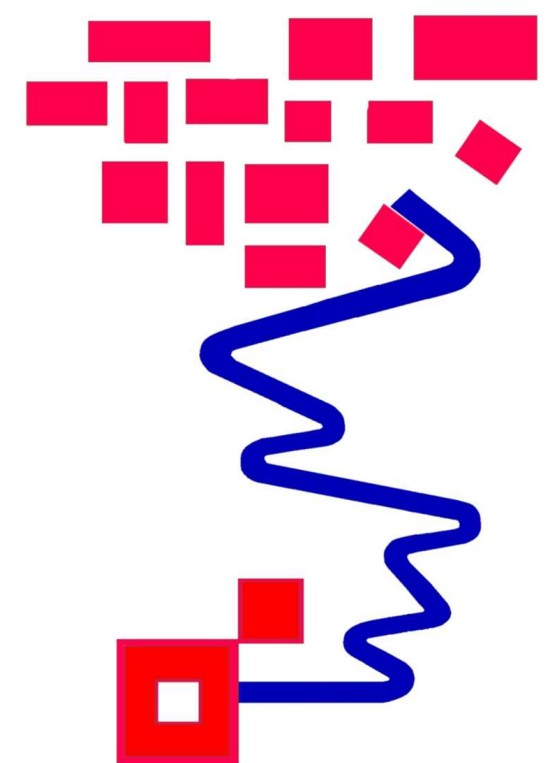


800                                             Fig. 07


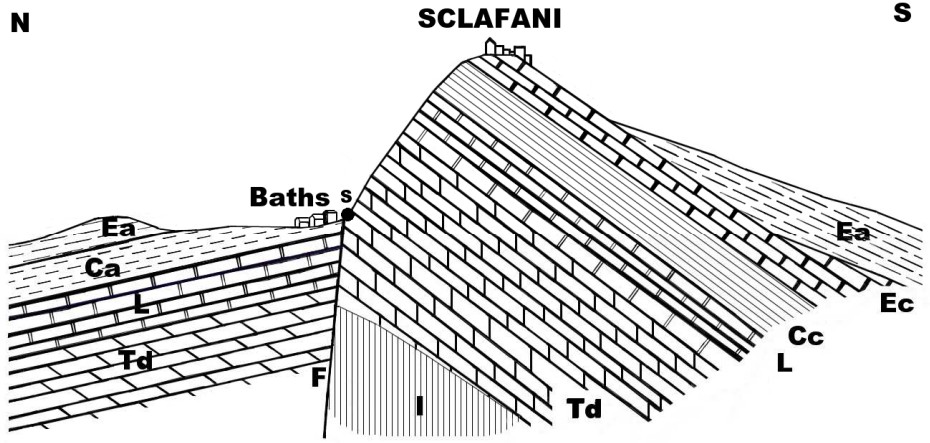

Fig. 08


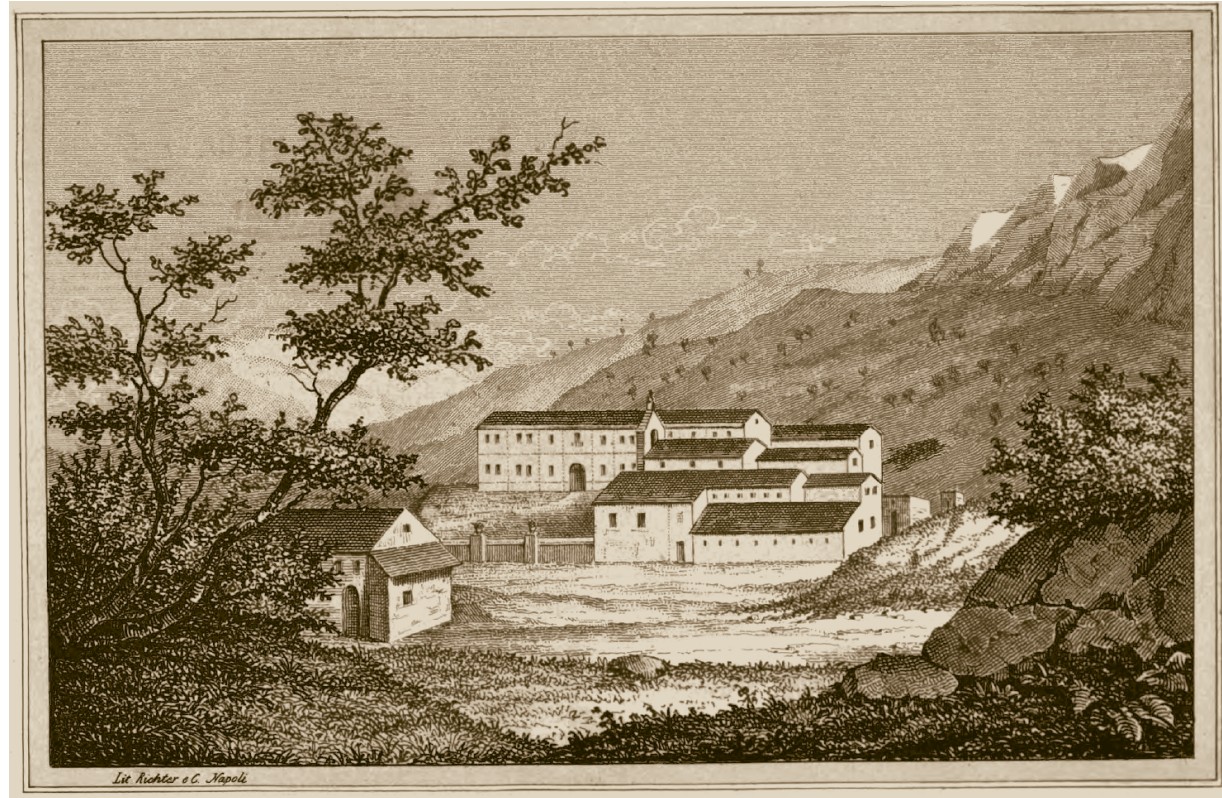

BAGNI TERMOMINERALI DI SCLAFANI


Fig. 09


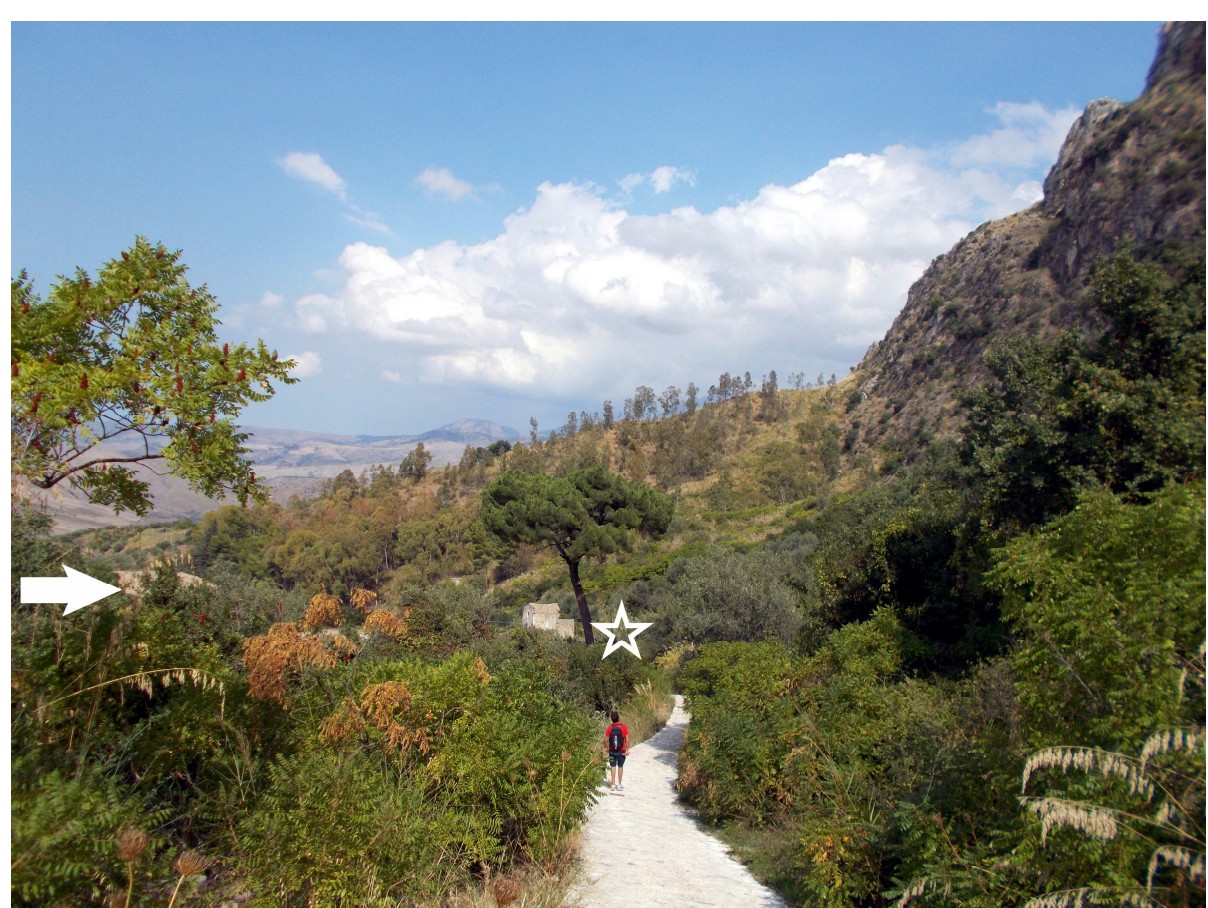


Fig. 10


