# Peer review of "Historical Analysis of Rainfall-Triggered Rockfalls: the Case Study of the Disaster of the Ancient Hydrothermal Sclafani Spa (Madonie Mts., Northern-Central Sicily, Italy) in 1851"

_Natural Hazards and Earth System Sciences, 2016_

## Referee Comment (RC1) · Anonymous Referee #1 · 10 Feb 2017

Dear authors thank you for your approach to reconstruct the circumstances of historic rockfalls. The procedures presented provide a valuable description on how to perform such an analysis.

Your title starts with "multidisciplinary approach to". In the article itself you did not go into detail of the multidisciplinarity. Therefore, I suggest to change the title to "Historical analysis of rainfall-triggered....".

P1L27-29: I do not see the relevance of this paragraph for the article and I would remove it. P1-2L30-44: Are these paragraphs relevant for the article? They are more or less a definition of landslide processes, aren't they? You could bring P2L45ff first and then explain the landslide definitions that they are later used in the article (are they?)

P2L70: "1.60m" above which level?

P7L228: "6,7" –> "6.7"

P8L252 ("understanding of the rockfall event") You did a nice analysis regarding the geology, landscape, the rainfall event and of the buildings. All based on a comprehensive literature research. The article title, however, promised information on "rockfalls". This would mean, mass involved, The event itself has not really been described yet. If possible, can you give some estimations on total height difference/horizontal distance/shadow angle/rock mass etc.?

P10L337: If more than 60000m2 are covered with accumulated rock material the event might not been classified as simple rockfall but a rockslide? What would you recommend?

---

## Referee Comment (RC2) · Anonymous Referee #2 · 24 Feb 2017

The authors should be acknowledged for their efforts in reconstructing the rockfall event. However, in my opinion, their work lacks of a significant scientific contribution and novelty.

The manuscript presents a summary of the historical documents describing the event. Contrary to the stated by the authors, the approach presented is not multidisciplinary as the results of the aerial photointerpretation and satellite images are not included. Both the geological and geomorphological contexts, including maps and figures, are

described at a scale too small for a proper appraisal of the predisposing factors in the slope and the development of the event.

No attempt is made to estimate the volume of the detached rock mass, the trajectories and extent of the deposits. The description of both predisposing and triggering factors is vague and not based on directly observed features in the rockfall source and other evidences. In fact, nothing is known about key features such as the rock mass strength, the joint pattern or the failure mechanism (p9, lines 280-283).

The conclusions do not reflect the content of the paper as the dynamics of the event has not been addressed and it is unlikely that the details provided could contribute to the quantification of the susceptibility of the slope to failure. The current stability conditions of the slope are not analyzed. Finally, I strongly disagree with the statement (p11, lines 370-371) on that the location, scale and frequency of rockfalls are unpredictable.

---

## Author Comment (AC1) · 27 Feb 2017

Dear Reviewer #1,

We are very grateful to you for expressing appreciation for our paper and providing us with useful suggestions and insightful comments. Below, you will find our answers to your careful suggestions, as well as changes made to our manuscript based on the corrections that you have recommended.

[Figure]

We are very glad to accept your precious suggestion to change the initial part of the title, because it places emphasis on the innovative historical approach developed in the paper.

P1L27-29: Text and respective references, removed.

P1-2L30-45 This part, introducing the topic of landslides, is not essential. We welcome your valuable suggestion to change its position in the paper (P2L45ff), because it improves its readability.

P2L70: "1.60 m above road level". Added.

P7L228: "6,7" –> "6.7". Corrected.

P8L252: Based on our estimations: total height difference (height of fall) is about 385 m; horizontal distance (length of runout) is about 572 m; shadow angle is about 31°-32°. Ratio of H/L = 0.67.

A reliable estimation of the rock volume deposit is very difficult, because no pre-event topographic map, to be compared with subsequent surveys (e.g., official maps of Italy, 1878), is available. The official cartography of the Bourbon Kingdom, "Map of the Palermo Region" (scale 1:20,000; equidistance: 18.52 m; original survey of the "Topographic Office" in Naples: 1849-52) originally included the Sclafani section. Unfortunately, this section is missing in the cartographic archives of the Italian Military Geographical Institute (Florence).

By using a new empirical relationship proposed by Guzzetti et al. (2009), which links the surface area to the volume of the landslide, we have attempted to estimate the rock volume, obtaining a value of about $6.8 \times 10^5$ m3. The same magnitude is obtained using the graph of volume versus ratio of H/L (Tianchi, 1983).

It is not possible to estimate the mass due to the heterogeneity of the deposit and the difficulty of determining the percentages of its constituent materials.

P10L337: Your question has been very enlightening. Undoubtedly, it is not easy to accurately classify a historical event that took place 150 years ago considering, among others, subsequent natural and anthropogenic changes (e. g., planting of tree species, terracing, excavations for road construction). The road built in 1930, whose excavation required the use of explosives, had a significant impact on the landscape, heavily changing its morphology, especially near the source area).

The event was a complex one; the type of initial failure evolved into another movement mechanism, when the material moved along the slope and changed its volume, incorporating materials entrained in its path. Indeed, in the kinematics of the event, the rockfall component cannot be ruled out, because the fragmented rock had to move beyond a break-away scarp (difference in height of about 70-90 m; topographical gradient of about 50°-60°, see fig. 05) at the lower cliff.

The accumulated material does not reflect the composition of the lithotypes outcropping in the source area (Ellipsactinia breccias), but rather the one of the rocks present in the entire slope (Ellipsactinia breccias, radiolarites, siliceous shales, marls, calcilutites, dolomites etc.).

Failing eyewitness reports, documentary data do not permit to easily classify the event. Synchronous documentary sources report the Italian term "scoscendimento", which at that time referred to a catastrophic landslide event, a veritable collapse of rock (see P10L324-327). In view of this, and considering that the surface covered by the accumulated material is significant, it is reasonable to suppose that the type of initial failure was a rockslide, probably a "rock collapse" (sensu Hungr and Evans, 2004).

Additional Literature

Hungr, O., Evans, S.G.: The occurrence and classification of massive rock slope failure, Felsbau, 22, 16–23, 2004.

Guzzetti, F., Ardizzone, F., Cardinali M., Rossi, M. and Valigi D.: Landslide volumes

and landslide mobilization rates in Umbria, central Italy, Earth and Planetary Science Letters, 279, 222–229, 2009.

Tianchi, L.: A mathematical model for predicting the extent of a major rockfall, Z. Geomorphol., 27 (4), 473-482, 1983

---

## Author Comment (AC2) · 9 Mar 2017

Dear Referee #2,

Our paper underlines the crucial importance of documentary data analysis to reconstruct the circumstances of landslide events that occurred in historical times, providing a significant methodological and scientific contribution of a pioneering nature.

We acknowledge your effort to identify the correct target of our paper. However, in

our opinion, your attempt has failed. Indeed, your comments lack an objective assessment of the fundamental role that historical datasets (documentary data, ancient maps, ancient engravings, etc.) play in the study of past landslide events. Your comments oversimplify and underestimate our archival contribution, reducing it to a mere "summary of the historical documents that describe the event". Our meticulous archival research work, with three documentary appendices (see Supplementary Information) including plenty of selected historical data, most of which unpublished (e.g. those from manuscript sources), was intended to offer a comprehensive analysis of historical sources in support of our assumptions and not just a list of collected data.

An example that can help clarify the mutual interaction between historical and geological data is the mapping of the landslide deposits from the Sclafani event. Geological and geomorphological evidence collected during field surveys, analysis of ancient maps, aerial and/or satellite images and historical data fit perfectly together, providing a detailed mapping of the area covered by the landslide deposits. We believe that there is no dichotomy between the data recorded in natural archives and those reported in historical archives: both are fundamental to the study of natural disasters. Our research rests upon the assumption: History for Earth Sciences, not History vs. Earth Sciences.

In recent times, Hungr (2004) stressed the importance of historical evidence, "potentially more accurate" than geological evidence (proxy data), even if "limited to the length of the historical period, often little more 100 years in much of the world". The catastrophic event of Sclafani, happened over 150 years ago, constitutes an interesting and emblematic case study.

Our historical reconstruction of the severe rainstorm of March 1851, and of the related Sclafani catastrophe, is supported by three different types of evidence. The first type is the direct description of the area of the thermal springs prior to the disaster by contemporary sources. These memories hold precious information about the landscape near the ancient thermal baths prior to the extreme event. The second type of documentary

source is represented by the records of local and regional authorities concerning measures taken to respond to the terrible disaster (destruction of thermal baths, water mills, roads etc.). A third source is the weather data kept by the Astronomic Observatory of the Palermo University (official) and by the Nautical Institute of Palermo (not official). We used these records to confirm the exact day of the disaster (previously incorrectly reported), as well as the impact and magnitude of the rainstorm, i.e. the main triggering factor. We consider that the manifold pieces of the Sclafani event puzzle, provided by documentary and geological evidence, fit entirely together, yielding a consistent picture of the impact of the disaster. The case study of Sclafani is an emblematic example that revives a catastrophic event ignored by the Italian inventories of landslide events (e.g. databases of ISPRA IFFI, AVI etc.).

The word "multidisciplinary" (in the title) was intended to highlight the dual contribution of different academic disciplines (Earth Sciences and History) to our research approach.

The "results of the aerial photointerpretation and satellite images" that, in your opinion, "are not included", are given in the map of Fig. 5, which outlines geological and geomorphological features (e.g. landslides) with a high degree of accuracy. The map is the synthesis of a detailed field survey, which was fine-tuned through a careful interpretation of topographical and cadastral maps, aerial photographs and satellite images.

For general assessments of geomorphological mapping in geohazards, Lee (2001) recommends 1:10,000 as a suitable scale. The map scale of 1:10,000 is the one of the Regional Technical Map of the Sicily Region. This is the scale chosen to build Italian official geological, geomorphological and hydrogeological maps (see ISPRA site, CARG Project). The area shown in the map is the minimum one that is required to describe the natural section outcropping in the environs of Sclafani.

Historical record collections do not include estimations of volume. A reliable estimation of volume and thickness is not possible, as no pre-event maps are available. With

regard to the area of the deposit, see P10L338. The exceptional rainfall event of March 1851, which devastated this north-western area of the Madonie mountains, must have certainly changed the lower talus slope (documentary sources report that the event caused an increase in ravines). As a result, any attempt to obtain a model of the possible trajectories related to the landslide would be unreliable. In addition, the soft rocks (radiolarites and siliceous shales), which form the lower talus slope, are prone to erosion; in 150 years, they certainly experienced denudation and modelling processes (above all during extreme rainfall events: 1886, 1890, 1895, 1919, 1925, 1929, 1931, 1954, 1964, 1976-77; 1985, see Aureli et al. 2008) making any model useless. Finally, the synchronous engraving (see Fig. 09), which represents the site of the ancient thermal spa, shows the vegetation cover of the talus; this vegetation is supposed to have had an impact on the trajectories of fall of the material. Unfortunately, Italian maps prior to the 20th century lack indications on vegetation covers.

The main triggering factor was the exceptional rainfall event of 12-13 March 1851. There is a cause-effect relationship between the exceptional rainstorm and the landslide, as substantiated by the numerous historical data that we retrieved. The area of Sclafani, typically mountainous, is subject to freeze-thaw conditions (see P9L286-289). The earthquake events that produced macroseismic effects in the study area in the first halves of the 19th century took place in 1818-19 and 1823 (Billi et al., 2010). Predisposing factors are many; some are intrinsic (related to the stratigraphic and tectonic setting), while other ones include selective erosion (hard-on-soft landforms, see P6L193-194; 199-200, 202-203). The anthropogenic impact changed the landscape near the source area (e.g. the road built in 1930, whose excavation required the use of explosives).

The conclusions show that geological and historical data fit reciprocally, making it possible to reconstruct a coherent picture of the event; a crucial role derives from the analysis of historical data that are the goals of the research carried out (see P11L357-362 and 366-368).

The event was a complex one; the type of initial failure evolved into another movement mechanism, when the material moved along the slope and changed its volume, incorporating materials entrained in its path. Indeed, the accumulated material does not reflect the composition of the lithotypes outcropping in the source area (Ellipsactinia breccias), but rather the one of the rocks present in the entire slope (Ellipsactinia breccias, radiolarites, siliceous shales, marls, calcilutites, dolomites etc.).

In the final part of your comments, you stated that: "it is unlikely that the details provided could contribute to the quantification of the susceptibility of the slope to failure". The data that we provided are propaedeutic. We never claimed that we could contribute "to quantifying" the susceptibility of the slope to failure (see P1L20-22). In P11L361-363, we merely reported the opinions of Authors (Porter and Orombelli, 1980; Wieczorek and Jäger, 1996) without comments. Finally, in the conclusions, with regard to susceptibility, we emphasise the need for conducting further investigations in order to gain more insight into our research findings (see P12L380-384).

As data on discontinuities are not available (see P9L280-283), no stability analysis is feasible.

In over 150 years, the lower talus slope certainly underwent erosion phenomena; therefore, its morphology cannot be regarded as constant in time; furthermore, empirical models are unable to predict the travel distance of future landslides (see Ayala-Carcedo et al., 2003) based on the data obtained for past events (e.g. the Sclafani catastrophe of March 1851).

Finally, in (P11L370-371), we merely quote the opinion of Zellmer (1987) without comments. We know that some researchers studied some possible precursors of rockfalls (mountain deformations: e.g. Bovis, 1990; seismic: Wang et al., 2003; Amitrano et al., 2005), including through monitoring systems (e.g. Schenato et al., 2013), in order to investigate the issue of prediction of these events, which are often catastrophic.

Additional References and Sitography

Amitrano D., Grasso, J.R., Senfaute G.: Seismic precursory patterns before a cliff collapse and critical point phenomena. Geophysical Research Letters 32: L08314, 2005.

Aureli, A., Contino, A., and Cusimano, G.: Aspetti idrogeologici e vulnerabilità all'Inquinamento degli acquiferi delle Madonie (Sicilia centro settentrionale). Note illustrative alla Carta della Vulnerabilità all'Inquinamento degli acquiferi delle Madonie (Sicilia centro settentrionale), Scala 1:50000. Regione Siciliana - Azienda Regionale Foreste Demaniali, Università degli Studi di Palermo, Dipartimento di Geologia e Geodesia, C. N. R.- G. N. D. C. I. - Pubbl. n. 2312, Collana Sicilia Foreste n. 39, Industria grafica Sarcuto, Agrigento, 168 pp., 2008.

AVI project.: avi.gndci.cnr.it/

Ayala-Carcedo, F.J., Cubillo-Nielsen, S., Alvarez, A., DomÄśnguez, M., LaÄśn, L., LaÄśn, R., and Ortiz, G.: Large Scale Rockfall Reach Susceptibility Maps in La Cabrera Sierra (Madrid) performed with GIS and Dynamic Analysis at 1:5000, Nat. Hazards, 30(3), 325–340, 2003.

Billi, A., Presti, D., Orecchio, B. Faccenna C. and Neri G.: Incipient extension along the active convergent margin of Nubia in Sicily, Italy: Cefalù-Etna seismic zone, Tectonics, 29, TC4026, doi:10.1029/2009TC002559, 2010.

Bovis, M.J.: Rock-slope deformation at Affliction Creek, southern Coast Mountains, British Columbia. Canadian Journal of Earth Sciences, 27, 243–254, 1990.

Hungr O.: Rock Avalanche Motion. Process and Modeling: Evans, S. G. and Martino S. (eds.), Massive Rock Slope Failure: New Models for hazard assessment, Celano, Italy, June 16-21, Abstract volume NATO and advanced Research Workshop, 66-69, 2002.

ISPRA CARG: http://www.isprambiente.gov.it/Media/carg/

ISPRA IFFI: http://www.isprambiente.gov.it/en/projects/soil-and-territory/iffiproject/default

Lee, E.M.: Geomorphological mapping, in J.S. Griffiths (ed.), Land Surface Evaluation for Engineering Practice, Geological Society Engineering Geology Special Publication 18, 53–56, 2001.

Schenato, L., Palmieri, L., Autizi, E., Calzavara, F., Vianello, L., Teza, G., Marcato, G., Sassi, R., Pasuto, A., Galgaro, A. and Galtarossa, A.: 'Rockfall precursor detection based on rock fracturing monitoring by means of optical fibre sensors', Int. J. Sustainable Materials and Structural Systems, Vol. 1, No. 2, pp.123–141, 2013.

Wang, W-N., Chigira, M., and Furuya, T.: Geological and geomorphological precursors of the Chiu-fen-erh-shan landslide triggered by the Chi-Chi earthquake in central Taiwan, Eng. Geol., 69, 1-13, 2003.

---

## Author Response (AR1)

Dear Editor,

We are very grateful to you for expressing appreciation for our answers to the comments made by the two reviewers. Thanks also for kindly inviting us to revise and resubmit our manuscript.

The paper has been modified taking into account the reviewers suggestions/comments and our own answers. Some changes have also been made to improve the quality and clarity of the text, as well as its general structure based on recommendations and the suggestions of the reviewers.

The corrections and/or changes are highlighted in yellow.

We would very much appreciate if you could consider our resubmission and we look forward to hearing your final decision as soon as possible.

Yours truly, Antonio Contino and co-authors

**RESPONSE TO THE FIRST REVIEWER**

**Comment from referee:** Dear authors thank you for your approach to reconstruct the circumstances of historic rockfalls. The procedures presented provide a valuable description on how to perform such an analysis.

**Author's response:** Dear Reviewer #1, We are very grateful to you for expressing appreciation for our paper and providing us with useful suggestions and insightful comments. Below, you will find our answers to your careful suggestions, as well as changes made to our manuscript based on the corrections that you have recommended.

**Comment from referee:**

Your title starts with "multidisciplinary approach to". In the article itself you did not go into detail of the multidisciplinarity. Therefore, I suggest to change the title to "Historical analysis of rainfall-triggered....".

**Author's response:**

We are very glad to accept your precious suggestion to change the initial part of the title, because it places emphasis on the innovative historical approach developed in the paper.

**Author's changes in manuscript:**

Initial part of the title (P1L1) as follows:

Historical analysis of.

**Comment from referee:**

P1L27-29: I do not see the relevance of this paragraph for the article and I would remove it.

**Author's response:**

P1L27-29: Text and respective references, removed.

**Author's changes in manuscript:**

Text in P1L27-29 (first version) and respective references (Walter, 2001, P17L559-560, first version), removed.

**Comment from referee:**

P1-2L30-44: Are these paragraphs relevant for the article? They are more or less a definition of landslide processes, aren't they? You could bring P2L45ff first and then explain the landslide definitions that they are later used in the article (are they?)

**Author's response:**

P1-2L30-45 This part, introducing the topic of landslides, is not essential. We welcome your valuable suggestion to change its position in the paper (P2L45ff), because it improves its readability.

**Author's changes in manuscript:**

Position of P2L45ff (first version), changed in P1L28-29.

**Comment from referee:**

P2L70: "1.60m" above which level?

**Author's response:**

P2L70: "1.60m above road level". Added.

**Author's changes in manuscript:**

"1.60m above road level". Added in P3L75.

**Comment from referee:**

P7L228: "6,7" –> "6.7"

**Author's response:**

P7L228: "6,7" –> "6.7". Corrected.

**Author's changes in manuscript:**

"6,7" –> "6.7". Corrected in P7L235.

**Comment from referee:**

P8L252 ("understanding of the rockfall event") You did a nice analysis regarding the geology, landscape, the rainfall event and of the buildings. All based on a comprehensive literature research. The article title, however, promised information on "rockfalls".

This would mean, mass involved, The event itself has not really been described yet.

If possible, can you give some estimations on total height difference/horizontal distance/ shadow angle/rock mass etc.?

**Author's response:**

P8L252: Based on our estimations: total height difference (height of fall) is about 385 m; horizontal distance (length of runout) is about 572 m; shadow angle is about 31°-32°. Ratio of H/L = 0.67.

A reliable estimation of the rock volume deposit is very difficult, because no pre-event topographic map, to be compared with subsequent surveys (e.g., official maps of Italy, 1878), is available. The official cartography of the Bourbon Kingdom, "Map of the Palermo Region" (scale 1:20,000; equidistance: 18.52 m; original survey of the "Topographic Office" in Naples: 1849-52) originally included the Sclafani section. Unfortunately, this section is missing in the cartographic archives of the Italian Military Geographical Institute (Florence).

By using a new empirical relationship proposed by Guzzetti et al. (2009), which links the surface area to the volume of the landslide, we have attempted to estimate the rock volume, obtaining a value of about $6.8 \times 10^5 \, m^3$. The same magnitude is obtained using the graph of volume versus ratio of H/L (Tianchi, 1983).

It is not possible to estimate the mass due to the heterogeneity of the deposit and the difficulty of determining the percentages of its constituent materials.

**Author's changes in manuscript:**

In page 12, line 377-390, as follows:

Some geometrical parameters of the Sclafani landslide could be determined: total height difference (height of fall) is about 385 m; horizontal distance (length of runout) is about 572 m; empirical shadow angle is about 31°-32°;  ratio of H/L = 0.67.

Historical record collections do not include estimations of volume. Hence, a reliable estimation of the rock volume deposit and thickness is very difficult, because no pre-event topographic map, to be compared with subsequent surveys (e.g., official maps of Italy, 1878), is available. The official cartography of the Bourbon Kingdom, "Map of the Palermo Region" (scale 1:20,000; equidistance: 18.52 m; original survey of the "Topographic Office" in Naples: 1849-52) originally included the Sclafani section. Unfortunately, this section is missing in the cartographic archives of the Italian Military Geographical Institute (Florence).

An attempt to estimate the rock volume by using a new empirical relationship proposed by Guzzetti et al. (2009), which links the surface area to the volume of the landslide. The resulting value was about $6.8 \times 10^5 \, m^3$. The same magnitude was obtained by using the volume vs. of H/L ratio graph (Tianchi, 1983).

Similarly, it was not possible to estimate the mass due to the heterogeneity of the deposit and the difficulty of determining the percentages of its constituent materials.

**Comment from referee:**

P10L337: If more than 60000m$^2$ are covered with accumulated rock material the event might not been classified as simple rockfall but a rockslide? What would you recommend?

**Author's response:**

P10L337: Your question has been very enlightening. Undoubtedly, it is not easy to accurately classify a historical event that took place 150 years ago considering, among others, subsequent natural and anthropogenic changes (e. g., planting of tree species, terracing, excavations for road construction). The road built in 1930, whose excavation required the use of explosives, had a significant impact on the landscape, heavily changing its morphology, especially near the source area).

The event was a complex one; the type of initial failure evolved into another movement mechanism, when the material moved along the slope and changed its volume, incorporating materials entrained in its path. Indeed, in the kinematics of the event, the rockfall component cannot be ruled out, because the fragmented rock had to move beyond a break-away scarp (difference in height of about 70-90 m; topographical gradient of about 50°-60°, see fig. 05) at the lower cliff.

The accumulated material does not reflect the composition of the lithotypes outcropping in the source area (Ellipsactinia breccias), but rather the one of the rocks present in the entire slope (Ellipsactinia breccias, radiolarites, siliceous shales, marls, calcilutites, dolomites etc.).

Failing eyewitness reports, documentary data do not permit to easily classify the event.

Synchronous documentary sources report the Italian term "scoscendimento", which at that time referred to a catastrophic landslide event, a veritable collapse of rock (see P10L324-327). In view of this, and considering that the surface covered by the accumulated material is significant, it is reasonable to suppose that the type of initial failure was a rockslide, probably a "rock collapse" (sensu Hungr and Evans, 2004).

**Author's changes in manuscript:**

In page 11 line 342-344, as follows:

Undoubtedly, failing eyewitness reports, documentary data do not permit to easily classify this historical disaster which took place over 150 years ago considering, among others, subsequent natural and anthropogenic changes (e.g., planting of tree species, terracing, excavations for road construction).

In page 11, line 346-348, as follows:

The event was a complex one: the type of initial failure evolved into another mechanism of movement, when the material advancing along the slope and changed its volume, by incorporating materials entrained in its path.

In page 11, line 349-351, as follows:

Indeed, in the kinematics of the event, the rockfall component cannot be ruled out, because the fragmented rock had to move beyond a break-away scarp (difference in height of about 70-90m; topographical gradient of about 50°-60°, see fig. 5) at the lower cliff.

In page 11, line 369-371, as follows:

Moreover, the accumulated material does not reflect the composition of the lithotypes outcropping in the source area (Ellipsactinia breccias), but rather the one of the rocks present in the entire slope (Ellipsactinia breccias, radiolarites, siliceous shales, marls, calcilutites, dolomites etc.).

In page 12, line 395-397, as follows:

In view of this and considering that the surface covered by the accumulated material and the estimated volume are significant, it is reasonable to suppose that the type of initial failure was a rockslide, probably a "rock collapse" (sensu Hungr and Evans, 2004).

In page 12, line 406, page 13, line 408, as follows:

This road, whose excavation required the use of explosives, had a significant impact on the landscape, heavily changing its morphology, especially near the source area.

**RESPONSE TO THE SECOND REVIEWER**

**Comment from referee:** The authors should be acknowledged for their efforts in reconstructing the rockfall event. However, in my opinion, their work lacks of a significant scientific contribution and novelty.

**Author's response:**

Dear Referee #2,

Our paper underlines the crucial importance of documentary data analysis to reconstruct the circumstances of landslide events that occurred in historical times, providing a significant methodological and scientific contribution of a pioneering nature.

We acknowledge your effort to identify the correct target of our paper. However, in our opinion, your attempt has failed. Indeed, your comments lack an objective assessment of the fundamental role that historical datasets (documentary data, ancient maps, ancient engravings, etc.) play in the study of past landslide events.

**Author's changes in manuscript:**

In page 2, line 59-61, as follows:

The paper underlines the crucial importance of documentary data analysis to reconstruct the circumstances of landslide events that occurred in historical times, providing a significant methodological and scientific contribution of a pioneering nature.

**Comment from referee:**

The manuscript presents a summary of the historical documents describing the event.

**Author's response:**

Your comments oversimplify and underestimate our archival contribution, reducing it to a mere "summary of the historical documents that describe the event". Our meticulous archival research work, with three documentary appendices (see Supplementary Information) including plenty of selected historical data, most of which unpublished (e.g. those from manuscript sources), was intended to offer a comprehensive analysis of historical sources in support of our assumptions and not just a list of collected data.

An example that can help clarify the mutual interaction between historical and geological data is the mapping of the landslide deposits from the Sclafani event. Geological and geomorphological evidence collected during field surveys, analysis of ancient maps, aerial and/or satellite images and historical data fit perfectly together, providing a detailed mapping of the area covered by the landslide deposits. We believe that there is no dichotomy between the data recorded in natural archives and those reported in historical archives: both are fundamental to the study of natural disasters. Our research rests upon the assumption: History for Earth Sciences, not History vs. Earth Sciences.

In recent times, Hungr (2004) stressed the importance of historical evidence, "potentially more accurate" than geological evidence (proxy data), even if "limited to the length of the historical period, often little more 100 years in much of the world". The catastrophic event of Sclafani, happened over 150 years ago, constitutes an interesting and emblematic case study.

Our historical reconstruction of the severe rainstorm of March 1851, and of the related Sclafani catastrophe, is supported by three different types of evidence. The first type is the direct description of the area of the thermal springs prior to the disaster by contemporary sources. These memories hold precious information about the landscape near the ancient thermal baths prior to the extreme event. The second type of documentary source is represented by the records of local and regional authorities concerning measures taken to respond to the terrible disaster (destruction of thermal baths, water mills, roads etc.). A third source is the weather data kept by the Astronomic Observatory of the Palermo University (official) and by the Nautical Institute of Palermo (not official). We used these records to confirm the exact day of the disaster (previously incorrectly reported), as well as the impact and magnitude of the rainstorm, i.e. the main triggering factor. We consider that the manifold pieces of the Sclafani event puzzle, provided by documentary and geological evidence, fit entirely together, yielding a consistent picture of the impact of the disaster. The case study of Sclafani is an emblematic example that revives a catastrophic event ignored by the Italian inventories of landslide events (e.g. databases of ISPRA IFFI, AVI etc.).

**Author's changes in manuscript:**

In page 12, line 374-376, as follows:

Geological and geomorphological evidence collected during field surveys, analyses of ancient maps, aerial and/or satellite images and historical data fit perfectly together, providing a detailed mapping of the area estimated (about 63,403 $m^2$) to be covered by the landslide deposits.

In page 2, line 55-58, as follows:

In recent times, Hungr (2004) stressed the importance of historical evidence, "potentially more accurate" than geological evidence (proxy data), even if "limited to the length of the historical period, often little more 100 years in much of the world". The catastrophic event of Sclafani, which happened over 150 years ago, constitutes an interesting and emblematic case study.

In page 8, line 246-253, as follows:

The historical reconstruction of the severe rainstorm of March 1851, and of the related Sclafani catastrophe, was supported by three different types of evidence: i) direct description of the area of the thermal springs prior to the disaster by contemporary sources; these memories hold precious information about the landscape near the ancient thermal baths prior to the extreme event; ii) records of local and regional authorities concerning measures taken to respond to the terrible disaster (destruction of thermal baths, water mills, roads etc.) and iii) weather data kept by the meteorological station of the OAP (official) and by the INP (non-official). These records made it possible to confirm the exact day of the disaster (previously incorrectly reported), as well as the impact and magnitude of the rainstorm, i.e. the main triggering factor.

In page 13, line 421-425, as follows:

The manifold pieces of the Sclafani event puzzle, provided by documentary and geological evidence, fit entirely together, yielding a consistent picture of the impact of the disaster. The analysis of historical data i.e. that are the goals of the research study played a crucial role. The case study of Sclafani is an emblematic example that revives a catastrophic event so far ignored by the Italian inventories of landslide events (e.g. databases of ISPRA IFFI, AVI etc.)

**Comment from referee:**

Contrary to the stated by the authors, the approach presented is not multidisciplinary…

**Author's response:**

The word "multidisciplinary" (in the title) was intended to highlight the dual contribution of different academic disciplines (Earth Sciences and History) to our research approach.

**Author's changes in manuscript:**

According with the recommendations of the reviewer 1#, Initial part of the title (P1L1) has been changed as it follows:

Historical analysis of.

**Comment from referee:**

….as the results of the aerial photointerpretation and satellite images are not included.

**Author's response:**

The "results of the aerial photointerpretation and satellite images" that, in your opinion, "are not included", are given in the map of Fig. 5, which outlines geological and geomorphological features (e.g. landslides) with a high degree of accuracy. The map is the synthesis of a detailed field survey, which was fine-tuned through a careful interpretation of topographical and cadastral maps, aerial photographs and satellite images.

**Author's changes in manuscript:**

In page 6, line 192-194, as follows:

The map of Fig. 5 is the synthesis of a meticulous field survey, which was fine-tuned by carefully interpreting of topographical and cadastral maps, aerial photographs and satellite images.

**Comment from referee:**

Both the geological and geomorphological contexts, including maps and figures, are described at a scale too small for a proper appraisal of the predisposing factors in the slope and the development of the event.

**Author's response:**

For general assessments of geomorphological mapping in geohazards, Lee (2001) recommends 1:10,000 as a suitable scale. The map scale of 1:10,000 is the one of the Regional Technical Map of the Sicily Region. This is the scale chosen to build Italian official geological, geomorphological and hydrogeological maps (see ISPRA site, CARG Project). The area shown in the map is the minimum one that is required to describe the natural section outcropping in the environs of Sclafani.

**Author's changes in manuscript:**

In page 6, line 191-192, as follows:

For general assessments of geomorphological mapping in geohazards, Lee (2001) recommends 1:10,000 as a suitable scale.

**Comment from referee:**

No attempt is made to estimate the volume of the detached rock mass,…

**Author's response:**

Historical record collections do not include estimations of volume. A reliable estimation of volume and thickness is not possible, as no pre-event maps are available.

**Author's changes in manuscript:**

In page 12, line 380, as follows:

Historical record collections do not include estimations of volume.

**Comment from referee:**

…..the trajectories and extent of the deposits.

**Author's response:**

With regard to the area of the deposit, see P10L338. The exceptional rainfall event of March 1851, which devastated this north-western area of the Madonie mountains, must have certainly changed the lower talus slope (documentary sources report that the event caused an increase in ravines). As a result, any attempt to obtain a model of the possible trajectories related to the landslide would be unreliable. In addition, the soft rocks (radiolarites and siliceous shales), which form the lower talus slope, are prone to erosion; in 150 years, they certainly experienced denudation and modelling processes (above all during extreme rainfall events: 1886, 1890, 1895, 1919, 1925, 1929, 1931, 1954, 1964, 1976-77; 1985, see Aureli et al. 2008) making any model useless. Finally, the synchronous engraving (see Fig. 09), which represents the site of the ancient thermal spa, shows the vegetation cover of the talus; this vegetation is supposed to have had an impact on the trajectories of fall of the material. Unfortunately, Italian maps prior to the 20th century lack indications on vegetation covers.

**Author's changes in manuscript:**

In page 11, line 359-368, as follows:

The exceptional rainfall event of March 1851, which devastated this north-western area of the Madonie mountains, must have certainly changed the lower talus slope (documentary sources report that the event caused an increase in ravines, see Supplementary Information, Table S1, source 14). As a result, any attempt to obtain a model of the possible trajectories related to the landslide would be unreliable. In addition, the soft rocks (radiolarites and siliceous shales), which form the lower talus slope, are prone to erosion; in 150 years, they certainly experienced denudation and modelling processes (above all during extreme rainfall events: 1886, 1890, 1895, 1919, 1925, 1929, 1931, 1954, 1964, 1976-77; 1985, see Aureli et al. 2008) thus making any model useless. The synchronous engraving (see Fig. 9), representing the site of the ancient thermal spa, shows the vegetation cover of the talus; this vegetation is supposed to have had an impact on the trajectories of fall of the material. Unfortunately, Italian maps prior to the $20^{th}$ century lack of reliable indications on vegetation covers.

**Comment from referee:**

The description of both predisposing and triggering factors is vague and not based on directly observed features in the rockfall source and other evidences. In fact, nothing is known about key features such as the rock mass strength, the joint pattern or the failure mechanism (p9, lines 280-283).

**Author's response:**

The main triggering factor was the exceptional rainfall event of 12-13 March 1851. There is a cause-effect relationship between the exceptional rainstorm and the landslide, as substantiated by the numerous historical data that we retrieved. The area of Sclafani, typically mountainous, is subject to freeze-thaw conditions (see P9L286-289). The earthquake events that produced macroseismic effects in the study area in the first halves of the 19th century took place in 1818-19 and 1823 (Billi et al., 2010).

Predisposing factors are many; some are intrinsic (related to the stratigraphic and tectonic setting), while other ones include selective erosion (hard-on-soft landforms, see P6L193-194; 199-200, 202-203). The anthropogenic impact changed the landscape near the source area (e.g. the road built in 1930, whose excavation required the use of explosives).

**Author's changes in manuscript:**

In page 13, line 437, page 14 line 445, as follows:

The main triggering factor of the Sclafani landslide was the exceptional rainfall event of 12-13 March 1851. There was a cause-effect relationship between the exceptional rainstorm and the landslide, as substantiated by the numerous historical data retrieved in this study (see Supplementary Information, Tables S2-S3). The area of Sclafani, typically mountainous, is subject to freeze-thaw conditions. The earthquake events that produced macroseismic effects in the study area in the first half of the 19[th] century took place in 1818-19 and 1823 (Billi et al., 2010). Predisposing factors were many; some were intrinsic (related to the stratigraphic and tectonic setting), while other ones included selective erosion (hard-on-soft landforms). The anthropogenic impact changed the landscape near the source area (e.g. the road built in 1930, whose excavation required the use of explosives).

**Comment from referee:**

The conclusions do not reflect the content of the paper as the dynamics of the event has not been addressed and it is unlikely that the details provided could contribute to the quantification of the susceptibility of the slope to failure. The current stability conditions of the slope are not analyzed.

**Author's response:**

The conclusions show that geological and historical data fit reciprocally, making it possible to reconstruct a coherent picture of the event; a crucial role derives from the analysis of historical data that are the goals of the research carried out (see P11L357-362 and 366-368).

The event was a complex one; the type of initial failure evolved into another movement mechanism, when the material moved along the slope and changed its volume, incorporating materials entrained in its path. Indeed, the accumulated material does not reflect the composition of the lithotypes outcropping in the source area (Ellipsactinia breccias), but rather the one of the rocks present in the entire slope (Ellipsactinia breccias, radiolarites, siliceous shales, marls, calcilutites, dolomites etc.).

In the final part of your comments, you stated that: "it is unlikely that the details provided could contribute to the quantification of the susceptibility of the slope to failure". The data that we provided

are propaedeutic. We never claimed that we could contribute "to quantifying" the susceptibility of the slope to failure (see P1L20-22). In P11L361-363, we merely reported the opinions of Authors (Porter and Orombelli, 1980; Wieczorek and Jäger, 1996) without comments. Finally, in the conclusions, with regard to susceptibility, we emphasise the need for conducting further investigations in order to gain more insight into our research findings (see P12L380-384).

As data on discontinuities are not available (see P9L280-283), no stability analysis is feasible.

In over 150 years, the lower talus slope certainly underwent erosion phenomena; therefore, its morphology cannot be regarded as constant in time; furthermore, empirical models are unable to predict the travel distance of future landslides (see Ayala-Carcedo et al., 2003) based on the data obtained for past events (e.g. the Sclafani catastrophe of March 1851).

**Author's changes in manuscript:**

In page 13, line 428-430, as follows:

According to some authors (e.g. Porter and Orombelli, 1980; Wieczorek and Jäger, 1996) detailed analyses of documentary data are crucial to identifying the mechanisms triggering rockfalls, evaluating the susceptibility of the various slopes to rockfalls and developing magnitude-frequency relationships.

In page 9, line 298-299, as follows:

Hence, given the lack of reliable discontinuity data, no stability analysis was feasible. This topic will be discussed in a future publication.

In page 11, line 371-373, as follows:

In addition, the morphology of the lower talus slope cannot be regarded as constant in time; therefore, empirical models are unable to predict the travel distance of future landslides (see Ayala-Carcedo et al., 2003) based on the data obtained for past events.

**Comment from referee:**

Finally, I strongly disagree with the statement (p11, lines 370-371) on that the location, scale and frequency of rockfalls are unpredictable.

**Author's response:**

Finally, in (P11L370-371), we merely quote the opinion of Zellmer (1987) without comments. We know that some researchers studied some possible precursors of rockfalls (mountain deformations: e.g. Bovis, 1990; seismic: Wang et al., 2003; Amitrano et al., 2005), including through monitoring systems (e.g. Schenato et al., 2013), in order to investigate the issue of prediction of these events, which are often catastrophic.

**Author's changes in manuscript:**

In page 14, line 446-450, as follows:

According to Zellmer (1987) the time, place and frequency of occurrence of rockfall disasters, as well as their scale, are unpredictable. However, some researchers are studying some possible precursors of rockfalls (mountain deformations: e.g. Bovis, 1990; seismic: Wang et al., 2003; Amitrano et al., 2005), including through monitoring systems (e.g. Schenato et al., 2013), in order to investigate the issue of prediction of these events, which are often catastrophic.

---

## Referee Report (RR1)

**Manuscript Review for nhess -2016-397, Historical Analysis of Rainfall -Triggered Rockfalls: the Case Study of the Disaster of the Ancient Hydrothermal Sclafani Spa (Madonie Mts., Northern-Central Sicily, Italy) in 1851 (authors: A. Contino et al.)**

Review 6 June 2017

**Scientific Significance. Does the manuscript represent a substantial contribution to the understanding of natural hazards and their consequences?**

**Rating: Good.**

The manuscript presents an interesting approach on how to reconstruct a hazardous historical rockfall event, by combining a detailed description of the geological setting and by tracing possible sources of information in historical records and existing literature. In such a way an important geo-hazard concerning the studied area, has been brought to the attention of the scientific community.

**Scientific Quality. Are the scientific and/or technical approaches valid?**

**Rating: Good.**

The authors have done a good job to collect all relevant information from historical sources. Generally the historical analysis of the event is of very good quality. The interpretation of the geological setting is also of good quality and informative to the reader. Furthermore, the production of the geomorphological map (presented in figure 5) as a result of detailed field mapping is of great importance for a correct reconstruction of the event.

By carefully reading the manuscript, I have gained a natural interest about the specific event and this has motivated a preliminary personal study on this specific site, mainly by collecting geographical information from internet resources (Google earth-maps etc.) Based on this study I would like to point out -suggest to the authors the following:

- It would have been great to include some kind of 3D information concerning the studied area. This could help the reader to get a better impression of the geological setting. It would also assist in a better geo-referencing of all exposed information in this research article. It might worth the effort to work in 3D and to produce a valid 3D model of the studied area. This would also enable a reconstruction of the event, by means of numerical modelling. The authors are correctly pointing out that geomorphological alterations in the area through time, make it difficult to model the rockfall event (i.e. rockfall trajectories), but my opinion is slightly different. A correct 3D model of the existing topography enriched with information concerning possible rockfall release positions and size of boulders (rockfall scenarios), could provide enough information for a preliminary dynamic analysis of the event by means of rockfall numerical modelling (in 2D or 3D). At least the 'Rockfall potential' of the given slope could be explored. This in turn, could yield information about the energy magnitude and the travel path of the historical event as well as for possible future events at the area. I have the feeling that the authors can greatly improve the manuscript by including such kind of information. The following figures come as a result of my personal study, driven by reading the manuscript and I am **only** including them in this review process, in order to make my points clearer to the authors:

[Figure]

*Figure 1: Study area, satellite imagery relative to figure 2 in the manuscript. Satellite imagery or ortho-photographs can assist in transmitting crucial information to the reader. In example, the structural geology (faults and other structural elements) of the studied area could be better explained with the aid of a proper ortho-photograph or satellite imagery. Of course, the image above and following images in this review, arrive from standard internet resources (Google maps).It might be possible to obtain satellite images of better quality from other sources.*

[Figure]

*Figure 2: Satellite imagery relative to the geomorphological map in figure 5. Geological formations could be better visualized on an ortho-photo or satellite imagery.*

[Figure]

*Figure 3: 3D overview of the area, assisting in a correct interpretation of geological structures. The cross section presented in figure 3 of the manuscript (Lower Cliff, Lower Talus Slope etc.) could be much better explained on the basis of a 3D model.*

[Figure]

*Figure 4: Georeferencing of data with the help of a 3D model?*

[Figure]

*Figure 5: Possible Rockfall scenarios? Identification of the position of the historical Sclafani Spa?*

My personal view is that there is not enough information in the manuscript that could enable a "dynamic-kinematic" reconstruction of the analysed event.

The description of the deposit and of the landform created as a consequence of the 1851 event could had been more detailed. The addition in the geomorphological map in figure 5, of the exact position of some silent witnesses (boulders transported by gravitational movements) concerning rockfall events at the same slope, could assist in quantifying the "slope dynamics" relative to rockfall events. A better description of size and sorting (there is only information about some very large boulders (in line 339: the largest ones are approximately equidimensional, 4 m in size?).This kind of additional information could assist towards a dynamic analysis of the event.

On the hydrological record concerning the studied area, is there enough data to allow the calculation of the recurrence interval (return period) of such powerful events as the described storm and the associated rainfall? This information could be usefull for risk calculations.

To conclude with: The historical analysis is of great quality and highlights how such an approach can provide with valuable information concerning geo-hazards. The geological description of the site and interpretation relative to the analyzed event is also of good quality. There is a lack of information that could enable a correct *dynamic* analysis of the event.

**Presentation Quality**

**Rating: Good.**

I believe the scientific data, results, and conclusions have been presented in a clear and well-structured way. I would have liked to see a properly scaled geological cross section similar to the one presented in figure 3, based on the geomorphological map in figure 5. A cross section indicating the event's travel path (i.e. from release position to the location of the ruined historical spa, according to the understanding of the authors) would in my opinion also improve the article. English language is properly being used throughout the manuscript, with limited typos, and the narrative style I have found it to be very interesting.

**For final publication:**

My opinion is that the manuscript should be accepted subject to minor revision.

---

## Author Response (AR2)

Dear Editor,

We are very grateful to you for kindly inviting us to revise and resubmit our manuscript.

The paper has been modified taking into account the reviewers' suggestions/comments and our own answers. The corrections and/or changes have been highlighted in yellow.

Yours truly,
Antonio Contino and co-authors.

**RESPONSE TO THE FIRST REVIEWER (REPORT 2)**

**Authors' response:** Dear Reviewer #1, we are very grateful to you for your very positive judgement.

**RESPONSE TO THE SECOND REVIEWER (REPORT 1)**

**Comment from referee:** The authors claim that their documentary data analysis to reconstruct the circumstances of landslide events occurring in historical times, is a significant methodological and scientific contribution of a pioneering nature. This is arguable, Geoarcheology is a well-established discipline. The geoarcheological approaches/techniques have been applied to the analysis of the occurrence of natural hazards on historical sites (e.g., Field & Banning, 1998 Geoarcheology, 13: 595-616; Bottari & Sepe, 2013: Quaternary Int. 309-309: 105-111). It is particularly well developed for the study of past earthquakes (e.g, Stiros, 2001-Jour. Structural Geology, 23: 545-562; Silva et al. 2005 Tectonophysics, 408: 129-146; Katz and Crouvi, 2007 Eng. Geology 95: 57-78; Rodriguez-Pascua et al. 2010 GSA Special Paper 471).

**Author's response:**
Dear Referee #2,

Our paper is based on the combined analysis of geological data and unpublished historical datasets (documentary data, ancient maps, ancient engravings, etc.), as well as of the existing relevant literature. This analysis has played a crucial role in this case study, concerning a landslide event that occurred in historical times. This is clearly inferred not only from the title of the paper, but also and above all from its para. 2.5 (Documentary Evidence).

We grant that geoarchaeology, which relates earth sciences to archaeology, has become a well-established discipline. However, the approach taken by and/or the techniques typically used by this disciplinary branch have no direct relevance to the content of our pioneering study (indeed, no reference is made to geoarchaeology in the abstract or body of our paper). It is only in the conclusions that we express the desirability of a future archaeological investigation in the study area (see 472-474). Finally, in all the relevant literature that you mentioned, no single paper used historical sources in support of research.

**Author's changes in manuscript:**

None.

**Comment from referee:**

In any case, the main criticism to the submitted manuscript is that the analysis of the failure mechanism, the sedimentological and/or textural description of the rockfall deposits and the geological characterization of the source (predisposing factors), which must be based on evidences and field observations, is missing. (...).

The mechanism is unclear. Despite the case is described as a rockfall, the authors conclude it was a rockslide (lines 391-397). This conclusion is based exclusively on the estimated volume of the deposits rather than on the analysis the kinematic features either observed at the deposits themselves or at the detachment area. Which is arguable.

The authors state (lines 361-362) that any attempt to obtain the possible trajectories related to the Sclafani landslide would be unreliable. Moreover (lines 372-375 and answers to the reviewers), they consider that the deposits might have been eroded. Taking this into account, how can the runout length, shadow, and travel distance angles and volume be determined?

**Author's response:**

The paper emphasises the complexity of the event (see lines 346-351) and the lithological heterogeneity of the landslide deposits, as well as of the entire slope, from the top of the castle to the bottom of the lower cliff (see lines 371-373).

We classified the event as a rockslide not only on the basis of the areal extent and estimated volume of the landslide deposits, but also and above all through cross-checks of mutually-fitting geological and historical data, thus providing a consistent picture in spite of the lack of eyewitness reports of an event that took place 160 years ago.

In the current stage of research, no pre- and post-event maps are available to make appropriate comparisons, especially in terms of morphology of the lower talus slope, whose dominant outcrops consist of soft rocks. Based on documentary sources, the extreme rainstorm of 12-13 Sept. 1851 triggered major erosional and landsliding events, which deeply and irreversibly changed many slopes, especially those made up of soft rocks, deepening and widening ravines (e.g. see Supplementary Information, Table S2, source 14). This suggests that the morphology of the lower talus slope may have been extremely different from the current one, prior to the triggering of the Sclafani landslide. Hence, any attempt to reconstruct the trajectories of the 1851 event from the current morphology of the lower talus slope may reasonably be poorly reliable or unreliable.

Finally, there has been a misunderstanding: erosional processes in the subsequent 160 years did not affect the landslide deposits, but rather the soft rocks (radiolarites and siliceous shales) forming the lower talus slope.

**Author's changes in manuscript:**

In page 11, lines 361-364, as follows:

This suggests that the morphology of the lower talus slope may have been extremely different from the current one, prior to the triggering of the Sclafani landslide. Hence, any attempt to reconstruct the trajectories of the 1851 event from the current morphology of the lower talus slope may reasonably be poorly reliable or unreliable.

**Comment from referee:**

An event attaining 0.68 million cubic meters that occurred 160 year ago, should have left a visible scar at the source. In fact, many Holocene rockfalls scarps of similar size are still observable in the landscape. Unfortunately, no description of the rockfall source is provided in the text. The statements on the role of the discontinuities in the instability (lines 270-276, 442-445) are not based on observations. A proper geomechanical or structural characterization of the rock cliff was not carried out. The authors mention (lines 296-299) that no published data are available about the level of fracturing of the bedrock and that slope is inaccessible. However, pictures S1 and S2 suggest otherwise. Rock mass outcrops may be accessed from castle and at the base of the cliff.

**Author's response:**

Unfortunately, historical sources fail to provide indications about the source area, even if they report that the source area was still active a few years after the 1851 event, as evidenced by further collapses of rock masses, including one (three years after the Sclafani landslide) that completely blocked the mule track connecting the built-up area with the thermal springs (see Supplementary Information, Table S2, source 7). This track rests entirely on the lower talus slope and is overhanged by the cliff of the castle, from which rocks continued to detach themselves. As mentioned in the paper, excavation works for building a road in 1930 radically changed the morphology of the area immediately overlying the source area, as use was made of explosives to cross the 1851 landslide rubble (now covered in part by vegetation).

To provide reliable data, a structural/geomechanical characterisation of the area should necessarily include in-depth surveys (particularly significant in the source area, where the detachment surface has an uneven pattern), possibly in 3D, of the entire rock body (Ellipsactinia breccias), i.e. of the source area of the 1851 event. In the future, this will be possible only by resorting to direct cliff-wall survey techniques of mountaineering/climbing type and/or to indirect survey techniques, including appropriate remote-sensing ones (e.g. terrestrial laser scanning). Moreover, at the site of the Sclafani castle, the top part of the rock outcrops is mostly masked by medieval fortifications.

**Author's changes in manuscript:**

None.

**Comment from referee:**

I invite the authors to have a look published works describing the deposits, the geological and the geomechanical contexts of ancient sites affected by landslides/rockfalls. These are someexamples: Dykes, 2007 (Landslides,4: 279-290); Senatore et al. 2013 (Geoarcheology, 29:1-15); Fanti et al. 2013 (Landslides, 10: 409-420); Gigli et al. 2012 (NHESS, 12:1883-1903); Zarroca et al. 2014 (Landslides, 11: 655-671); Margottini et al. 2015 (Landslides, 12: 193-204); Gül et al. 2016 (Environ. Earth Sci. 75: 1310)

**Author's response:**

Dear Referee #2, we thank you for quoting these interesting papers about ancient landslide sites, although their geological and geomorphological settings have very few points of similarity with our case study. However, none of these studies uses historical data from documentary sources and/or historical maps in support of research.

**Author's changes in manuscript:**

None.

**RESPONSE TO THE SECOND REVIEWER (REPORT 3)**

**Author's response:**

Dear Reviewer #3,

We are very grateful to you for expressing appreciation for our paper and providing us with useful suggestions and insightful comments. Below, you will find our answers to your careful suggestions, as well as the changes made to our manuscript that you have recommended.

**Comment from referee:**

- It would have been great to include some kind of 3D information concerning the studied area. This could help the reader to get a better impression of the geological setting. It would also assist in a better geo-referencing of all exposed information in this research article. It might worth the effort to work in 3D and to produce a valid 3D model of the studied area. This would also enable a reconstruction of the event, by means of numerical modelling. The authors are correctly pointing out that geomorphological alterations in the area through time, make it difficult to model the rockfall event (i.e. rockfall trajectories), but my opinion is slightly different. A correct 3D model of the existing topography enriched with information concerning possible rockfall release positions and size of boulders (rockfall scenarios), could provide enough information for a preliminary dynamic analysis of the event by means of rockfall numerical modelling (in 2D or 3D). At least the 'Rockfall potential' of the given slope could be explored. This in turn, could yield information about the energy magnitude and the

travel path of the historical event as well as for possible future events at the area. I have the feeling that the authors can greatly improve the manuscript by including such kind of information.(….)

Figure 1: Study area, satellite imagery relative to figure 2 in the manuscript. Satellite imagery or ortho-photographs can assist in transmitting crucial information to the reader. In example, the structural geology (faults and other structural elements) of the studied area could be better explained with the aid of a proper ortho-photograph or satellite imagery. Of course, the image above and following images in this review, arrive from standard internet resources (Google maps). It might be possible to obtain satellite images of better quality from other sources.

Figure 2: Satellite imagery relative to the geomorphological map in figure 5. Geological formations could be better visualized on an orthophoto or satellite imagery.

Figure 3: 3D overview of the area, assisting in a correct interpretation of geological structures. The cross section presented in figure 3 of the manuscript (Lower Cliff, Lower Talus Slope etc.) could be much better explained on the basis of a 3D model.

Figure 4: Georeferencing of data with the help of a 3D model?

Figure 5: Possible Rockfall scenarios? Identification of the position of the historical Sclafani Spa?

**Author's response:**

We feel indebted to you for your valuable suggestions, especially because they provide a significant contribution to enhancing the clarity and thus quality of our paper. We have accepted your suggestion to add a 3D view of the surrounding area to the Supplementary Information, specifying the location of the site of the ancient spa and of the thermal springs, as well as the vantage point from which the photo in fig. 10 was taken.

**Author's changes in manuscript:**

See figure S5 in the Supplementary Information.

**Comment from referee:**

My personal view is that there is not enough information in the manuscript that could enable a "dynamic-kinematic" reconstruction of the analysed event. The description of the deposit and of the landform created as a consequence of the 1851 event could had been more detailed. The addition in the geomorphological map in figure 5, of the exact position of some silent witnesses (boulders transported by gravitational movements) concerning rockfall events at the same slope, could assist in quantifying the "slope dynamics" relative to rockfall events. A better description of size and sorting (there is only information about some very large boulders (in line 339: the largest ones are approximately equidimensional, 4 m in size?)

**Author's response:**

The mapping of some silent witnesses could be very useful, but it would require additional in-depth field surveys.

The materials making up the landslide deposits are of variable size: very small for fragments of siliceous shales and/or radiolarites (cm/dm), progressively larger from calcilutites (mostly dm, occasionally up to 1 m) to dolomites and limestones (from dm to some m). In the field, naturally-exposed landslide deposits are often obliterated by a thick vegetal cover. Small partial exposures, resulting from earthworks, show carbonate blocks (limestones and dolomites), generally of metre scale, embedded into a chaotic mass of fragments of radiolarites, siliceous shales and slab-shaped calcilutites.

**Author's changes in manuscript:**

In pages 11-12, lines 373-378, as follows:

The materials making up the landslide deposits are of variable size: very small for fragments of siliceous shales and/or radiolarites (cm/dm), progressively larger from calcilutites (mostly dm, occasionally up to 1 m) to dolomites and limestones (from dm to some m). In the field, naturally-exposed landslide deposits are often obliterated by a thick vegetal cover. Small partial exposures, resulting from earthworks, show carbonate blocks (limestones and dolomites), generally of metre scale, embedded into a chaotic mass of fragments of radiolarites, siliceous shales and slab-shaped calcilutites.

**Comment from referee:**

On the hydrological record concerning the studied area, is there enough data to allow the calculation of the recurrence interval (return period) of such powerful events as the described storm and the associated rainfall? This information could be useful for risk calculations.

**Author's response:**

Unfortunately, the relevant data are insufficient. So far, the statistical analyses of the hydrological records for central-western Sicily (the time series of the Palermo astronomical observatory, cfr. Micela, G. et al., *Due secoli di pioggia a Palermo*, Osservatorio Astronomico di Palermo G. S. Vaiana, Università di Palermo, Palermo 2001, 292 pp.) have not yielded meaningful results (e.g. cfr. Iuliano, V. and Nastasi, P., 1985, *Prime considerazioni sulla distribuzione statistica dei totali annui di Pioggia per Palermo (Osservatorio Astronomico)*, Rivista di Meteorologia Aeronautica, vol. 45, 2-3, pp. 91-99). A detailed investigation would certainly be helpful, i.e. by reviewing the hourly diagrams and the records of the Palermo astronomical observatory; indeed, in spite of their deficiencies, these data are only source available that covers a sufficiently long period of time (since 1801).

As to the Madonie mountains, hydrological records are very poor (cfr. Aureli, A., Contino, A. and Cusimano, G., *Aspetti idrogeologici e vulnerabilità all'inquinamento degli acquiferi delle Madonie (Sicilia centro settentrionale)*, Regione Siciliana Azienda Regionale Foreste Demaniali, Università degli Studi di Palermo, Dipartimento di Geologia e Geodesia, Consiglio Nazionale delle Ricerche, Gruppo Nazionale per la Difesa dalle Catastrofi Idrogeologiche, Pubbl. n. 2312, Collana Sicilia Foreste, 39, 168 pp., Sarcuto, Agrigento, 2008). Temperature and rain gauge stations in the area are decentralised with respect to the main mountain peaks and lie at much lower elevations. The few existing hydrometric stations have worked for too short periods and with numerous interruptions.

**Author's changes in manuscript:**

None.

**Comment from referee:**

I would have liked to see a properly scaled geological cross section similar to the one presented in figure 3, based on the geomorphological map in figure 5. A cross section indicating the event's travel path (i.e. from release position to the location of the ruined historical spa, according to the understanding of the authors) would in my opinion also improve the article.

**Author's response:**

We feel indebted to you for your valuable suggestions, especially because they provide a significant contribution to enhancing the clarity and thus quality of our paper.

**Author's changes manuscript:**

See figure S6 in the Supplementary Information.